



**Seasonal characteristics of atmospheric peroxyacetyl nitrate (PAN) in a coastal city**
**of Southeast China: Explanatory factors and photochemical effects**
Taotao Liu[1,2,3], Gaojie Chen[1,2,3], Jinsheng Chen[1,2]*, Lingling Xu[1,2], Mengren Li[1,2], Youwei Hong[1,2]*, Yanting Chen[1,2],
Xiaoting Ji[1,2,3], Chen Yang[1,2,3], Yuping Chen[1,2,3], Weiguo Huang[4], Quanjia Huang[5], Hong Wang[6]
[1]Center for Excellence in Regional Atmospheric Environment, Institute of Urban Environment, Chinese Academy of Sciences,
Xiamen, China
[2]Key Lab of Urban Environment and Health, Institute of Urban Environment, Chinese Academy of Sciences, Xiamen, China
[3]University of Chinese Academy of Sciences, Beijing, China
[4]State Key Laboratory of Structural Chemistry, Fujian Institute of Research on the Structure of Matter, Chinese Academy of
Sciences, Fuzhou, China.
[5]Xiamen Environmental Monitoring Station, Xiamen, China
[6]Fujian Meteorological Science Institute, Fujian Key Laboratory of Severe Weather, Fuzhou, China
Corresponding authors E-mail: Jinsheng Chen (jschen@iue.ac.cn); Youwei Hong (ywhong@iue.ac.cn)
**Abstract:**

19        Peroxyacetyl nitrate (PAN) acting as a typical indicator of photochemical pollution can redistribute

NOx and modulate $O_3$ production. Coupled with the observation-based model (OBM) and a generalized
additive model (GAM), the intensive observation campaigns were conducted to reveal the pollution
characteristics of PAN and its impact on $O_3$, the contributions of influencing factors to PAN formation
were also quantified in this paper. The F-values of GAM results reflecting the importance of the
influencing factors showed that ultraviolet radiation (UV, F-value=60.64), Ox (Ox=$NO_2$+$O_3$, 57.65), and
air temperature (T, 17.55) were the main contributors in the PAN pollution in spring, while the significant
effects of Ox (58.45), total VOCs (TVOCs, 21.63) and T (20.46) were found in autumn. The PAN
formation rate in autumn was 1.58 times higher than that in spring, relating to the intense photochemical
reaction and meteorological conditions. Without considering the transformation of peroxyacetyl radical
(PA) and PAN, acetaldehyde contributed to the dominant production of PA (46±4%), followed by
methylglyoxal (28±3%) and radical cycling (19±3%). The PAN formation was highly VOC-sensitive, and
sufficient NOx (compared with VOCs abundance) would not be the limited factor for atmospheric
photochemistry. PAN could promote or inhibit $O_3$ formation under high or low ROx levels, respectively.
The PAN promoting $O_3$ formation mainly occurred during the periods of 11:00-16:00 (local time) when
the favorable meteorological conditions (high UV and T) stimulated the photochemical reactions to offer
ROx radicals, which accounted for 17% of the whole monitoring periods in spring and 31% in autumn.
In this study, the formation mechanism of PAN and its effect on ozone were identified, which might be





helpful to improve the scientific understanding of photochemical pollution in coastal areas.

**Keywords**: PAN formation mechanism; GAM model; OBM-MCM; Sensitivity analysis; Photochemical
pollution; Coastal area


**1 Introduction**

Peroxyacetyl nitrate ($CH_3C(O)O_2NO_2$, PAN) is a key product of photochemical smog (Penkett and

Brice, 1986; Li et al., 2019). PAN is generated through photochemical reactions of precursors emitted by
human activities only, and the atmospheric PAN is a reliable and scientific indicator of photochemical
pollution (Lonneman et al., 1976; Han et al., 2017). In the surface atmosphere, the level of PAN is much
lower than that of ozone ($O_3$), but its biological toxicity is about one or two magnitudes greater than that
of $O_3$ (Temple and Taylor, 1983). Additionally, PAN acts as a temporary reservoir for NOx and radicals,
and can transport to remote regions to redistribute NOx and intervene in $O_3$ production at regional or even
global scale (Kleindienst, 1994; Atkinson et al., 2006; Fischer et al., 2010).

The reaction of peroxyacetyl radical ($CH_3C(O)O_2$, PA) with $NO_2$ is solely formation pathway of

PAN (Han et al., 2017; Xue et al., 2014). PAN affects radical chemistry and modulates $O_3$ production
mainly by affecting PA radical, which is one of the most abundant organic peroxy radicals in the
troposphere (Tyndall et al., 2001). Only a small group of oxygenated volatile organic compounds (OVOCs)
(i.e. acetaldehyde ($CH_3CHO$), methacrolein (MACR), methyl vinyl ketone (MVK), methyl ethyl ketone
(MEK), and methylglyoxal (MGLY)) can directly produce PA radical to generate PAN (Xue et al., 2014;
Zhang et al., 2015). These OVOCs (the second-generation precursors of PAN) are mainly transformed by
oxidation reactions from some hydrocarbons such as ethane, propene, isoprene and aromatics (the first-
generation precursors of PAN) (Xu et al., 2021). The main and direct PAN destruction is thermal
decomposition, and the indirect sinks of PAN were the reactions of PA with NO, $HO_2$, and $RO_2$ (Wolfe et
al., 2014; Zeng et al., 2019).

Some studies on the distribution and sources of PAN have been conducted in urban, suburban, and

remote regions around the world (Grosjean et al., 2002; Marley et al., 2007; Roberts et al., 2001). The
mixing ratio of PAN in cities is higher than that in rural and remote areas, and that in background areas
such as oceans and mountains can be as low as tens of pptv (Gaffney et al., 1999; Moore et al., 2009).
Despite the growing concerns about photochemical pollution in China, PAN measurements and analysis
of its photochemical mechanism are still sparse (Zeng et al., 2019). At present, the observations of PAN





were mainly distributed in Beijing, Guangzhou, and Hong Kong (Xue et al., 2014; Yuan et al., 2018; Zeng
et al., 2019). Xue et al. (2014) reported that anthropogenic VOCs were the most important precursors of
PAN in urban areas, and isoprene was the predominant precursor in suburban regions. In Zeng et al. (2019)
study, carbonyls offered the highest contribution to PAN production, followed by aromatics and BVOCs.
In addition, some researchers found that atmospheric PAN suppressed local $O_3$ formation in autumn (Zeng
et al., 2019). Recently, negative and positive impacts of PAN photochemistry on $O_3$ production were
captured under the low and high NOx conditions, respectively. However, the PAN formation and its
influencing mechanism on $O_3$ production are still complex and unclear (Hu et al., 2020; Zhang et al 2019;
Xu et al., 2018). Long-term field measurements and model simulations could help to verify the
mechanisms under various pollution scenarios and environmental conditions.

Xiamen, with meteorological conditions of strong radiation, high relative humidity and air

temperature, is located in the coastal region of Southeast China (Liu et al., 2020a; Liu et al., 2020b). Rapid
urbanization has caused photochemical pollution in the coastal city, with relatively high levels of nitrogen
oxides and volatile organic compounds. Our previous studies focused on the occurrence and pollution
characteristics of PAN (Hu et al., 2020). In this study, an observation-based model coupled to the Master
Chemical Mechanism (OBM-MCM) was used to better understand PAN photochemistry, and a
generalized additive model (GAM) was adopted to quantify the complex nonlinear relationships of PAN
with its precursors and environmental factors (Hua et al., 2021). The study aims to explore (1) the PAN
formation mechanism and sensitivity analysis, (2) the impacts of PAN on $O_3$ formation and radical
chemistry, (3) the relationship between PAN and influencing factors under different pollution scenarios.

**2 Materials and methods**
**2.1 Study site and observations**

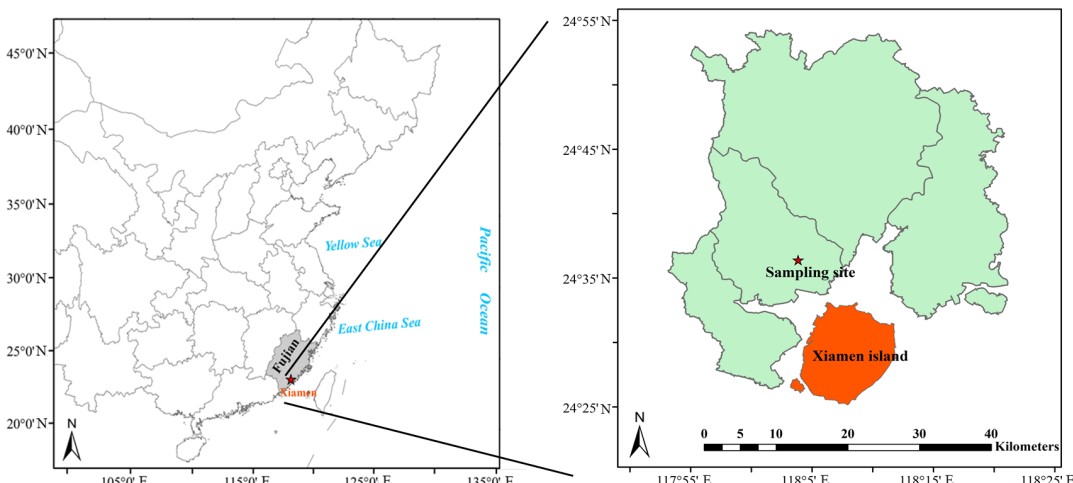

**Fig. 1. Location of Xiamen and the observation site.**

Observations were carried out at the Atmospheric Environment Observation Supersite (AEOS, 24.61° N, 118.06° E; Fig. 1), located on the rooftop of around a 70 m high building in the Institute of Urban Environment, Chinese Academy of Sciences. The observations site is surrounded by highways, educational institutions, and residential buildings, which was characterized by rapidly urbanizing development area. When the prevailing wind direction was southerly winds, our observation site was attributed to the downwind region of the downtown (Xiamen island) with densely population and heavy traffic (Hu et al., 2020; Liu et al., 2021a). The field observation was continuously conducted from March 15 to November 4, 2020. The photochemical pollution events mainly appeared during spring and autumn in Xiamen, and we preferred to choose the periods with relatively high $O_3$ and PAN levels, then the measured data of 53 days in each season was chosen after excluding some special circumstances, such as extreme synoptic situations and instrument calibration. Xiamen is under the East Asian monsoon control, belonging to the subtropical marine climate. In spring, north cold airflow and south warm airflow formed the quasistationary front causing atmospheric stagnation. In autumn, under the control of the west pacific subtropical high (WPSH), high T and low RH enhanced the formation and accumulation of photochemical pollutants (Wu et al., 2020).

PAN was monitored using a PAN analyzer (PANs-1000, Focused Photonics Inc., Hangzhou, CN) containing gas chromatography with electron capture detector (GC-ECD). During the observation period, multi-point standard curve calibration was conducted once a month, and single-point calibration was conducted every week, respectively. In the calibration mode of the PAN analyzer, the Mass Flow Controller (MFC) controls the flow rate of NO, acetone and zero gas separately. The PAN standard gas is



generated by the reaction of NO and acetone under ultraviolet light irradiation, and the sample is diluted
to the required calibration mixing ratio for injection analysis. PAN was detected every 5 min and the
detection limit was 50 pptv. The uncertainty and precision of PAN measurement were ±10% and 3%,
respectively. Criteria air pollutants of $O_3$, CO, $SO_2$, and NOx, were monitored by using Thermo
Instruments TEI 49i, 48i, 43i, and 42i (Thermo Fisher Scientific, Waltham, MA, USA), respectively.
Particulate matters ($PM_{2.5}$) were monitored by oscillating microbalance with tapered element
(TEOM1405, Thermo Scientific Corp., MA, US), and the uncertainty of the $PM_{2.5}$ measurement was
±20%, respectively. The meteorological parameters (i.e. wind speed (WS), wind direction (WD), pressure
(P), air temperature (T), and relative humidity (RH)) were measured by an ultrasonic atmospherium
(150WX, Airmar, USA). Ultraviolet radiation (UV) was determined by a UV radiometer (KIPP &
ZONEN, SUV5 Smart UV Radiometer). HONO was monitored using an analyzer for Monitoring
Aerosols and Gases in Ambient Air (MARGA, ADI 2080, Applikon Analytical B.V., the Netherlands). A
gas chromatography-mass spectrometer (GC-FID/MS, TH-300B, Wuhan, CN) was used for monitoring
the atmospheric VOCs with a 1-hour time resolution. The single-point calibration was performed every
day at 23:00 with the standard mixtures of PAMS and TO15, and multi-point calibration was performed
one month. The detection limits of the measured VOCs were in the range of 0.02 ppbv to 0.30 ppbv, and
the measurement precision was ≤10%. Photolysis frequencies including $J(O^1D)$, $J(NO_2)$, $J(HONO)$,
$J(NO_3)$, $J(HCHO)$, and $J(H_2O_2)$ were analyzed by a photolysis spectrometer (PFS-100, Focused
Photonics Inc., Hangzhou, China), and the uncertainty and detection limit of photolysis rates
measurement were ±5% and around $1×10^{-5}$, respectively. Table S1 shows the detailed uncertainty and
detection limit of instruments for trace gas observation. A schedule was applied to operate and inspect the
AEOS monitoring station regularly and strictly to ensure the validity of the data. The detailed applications
of the atmospheric monitoring procedure were shown in our previous studies (Wu et al., 2020; Liu et al.,
2020a; Liu et al., 2020b; Hu et al., 2020).

### 2.2 Observation-based model

The OBM-MCM model is successfully used in the simulation of photochemical processes and the

quantification of the reaction rates, such as $O_3$, PAN and alkyl nitrates ($RONO_2$) (Zeng et al. 2019). In
our study, the PAN photochemistry mechanism was simulated using this box model, and the incorporated
chemical mechanism was the latest version of MCM-v3.3.1 (http://mcm.leeds.ac.uk/MCM/), which
introduced 142 nonmethane VOCs and about 20000 elementary reactions (Jenkin et al., 2003; Saunders





et al., 2003). The physical process including dilution effect and dry deposition within the boundary layer
height was considered, avoiding the excessive accumulation of pollutants in the model (Li et al., 2018;
Liu et al., 2021; Xue et al., 2016). The observed data with a time resolution of 1 h of pollutants,
meteorological parameters, and photolysis rate constants, which were mentioned in Section 2.1, were
input into the OBM-MCM model as constraints. The photolysis rates of other molecules were driven by
solar zenith angle and were scaled by measured $JNO_2$ (Saunders et al., 2003). Pre-ran for 2 days before
running the model to constrain the unmeasured compounds reaching a steady-state (Xue et al., 2014). The
detailed model introduction showed in our previous study (Liu et al., 2021a).
PAN affects atmospheric photochemistry by acting as a temporary source or sinks of PA radical (Xue
et al., 2014; Liu et al., 2021), hence the production and sink of PA radical reflecting the PAN formation
were discussed in our study. Furthermore, relative incremental reactivity (RIR) was used to analyze the
sensitivity of $O_3$ (Eq. 1) and PAN (Eq. 2) to their precursors, and was calculated as the ratio of the
differences in $O_3$ or PAN net production rate to variety in precursors (Chen et al., 2020; Liu et al., 2021).
The detailed net production rate of $O_3$ ($P(O_3)$) was introduced in our previous study (Liu et al., 2021a).
The net production of PAN ($P(PAN)$) involved the production pathway of $PA+NO_2$, and the loss of PAN
was thermal decomposition and $PAN+OH$ (Zeng et al., 2019).

$$RIR(PAN) = \frac{\Delta P(O_3)/P(O_3)}{\Delta X/X} \qquad (1)$$

$$RIR(O_3) = \frac{\Delta P(PAN)/P(PAN)}{\Delta X/X} \qquad (2)$$

Here, the $\Delta X/X$ meaning the reduction in the input mixing ratios of each target $O_3$ and PAN precursor
group was 20% (Liu et al., 2021).

**2.3 Generalized additive model**

The Generalized Additive Model (GAM) is an extension of the additive model proposed. Different
from traditional regression models, GAM is a non-parametric regression model driven by data rather than
statistical distribution models (He et al., 2017). GAM does not need to set the parameter model in advance,
and it can adjust the functional form of the explained variable according to the specific situation. The
Generalized Additive Model (GAM) has been widely used in air pollution research such as $O_3$ and $PM_{2.5}$,
and can effectively deal with the complex nonlinear relationship between air pollutants and influencing
factors (Ma et al., 2020; Hua et al., 2021; Guan et al., 2019). It is the first time that the GAM is used to
analyze the relationship between PAN and its influencing factors, and the combined effect of multiple
influencing factors on the PAN mixing ratio was discussed in our study. Its form is:





$$g(y)=\beta+f_1(x_1)+f_2(x_2)+\ldots\ldots+f_n(x_n)+\alpha \qquad (3)$$
Where $y$ is the response variable; $g(y)$ is the connection function; $x_n$, $x_i$, $x_j$, $x_k$, and $x_l$ are the
explanatory variables; fn is the non-parametric smoothing functions; $\beta$ is the intercept; $\alpha$ is the truncation
error.
The F-value, P-value, adjust $R^2$, and deviance explained given by the GAMs model are used to judge
the significance of the influencing factors on PAN and the goodness of the model simulation. Among
them, a high F-value indicates the great importance of the influencing factor; the P-value is used to judge
the significance of the model result; the adjusted $R^2$ is the value of the regression square ranging from 0
to 1; the deviance explained represents the fitting effect. In addition, when the degree of freedom (edf,
ref.df) of the explanatory variable is 1, it indicates that the explanatory variable and the response variable
are linear. When the degree>1, it is a non-linear relationship.

**3 Results and discussion**
**3.1. Overview of observation**

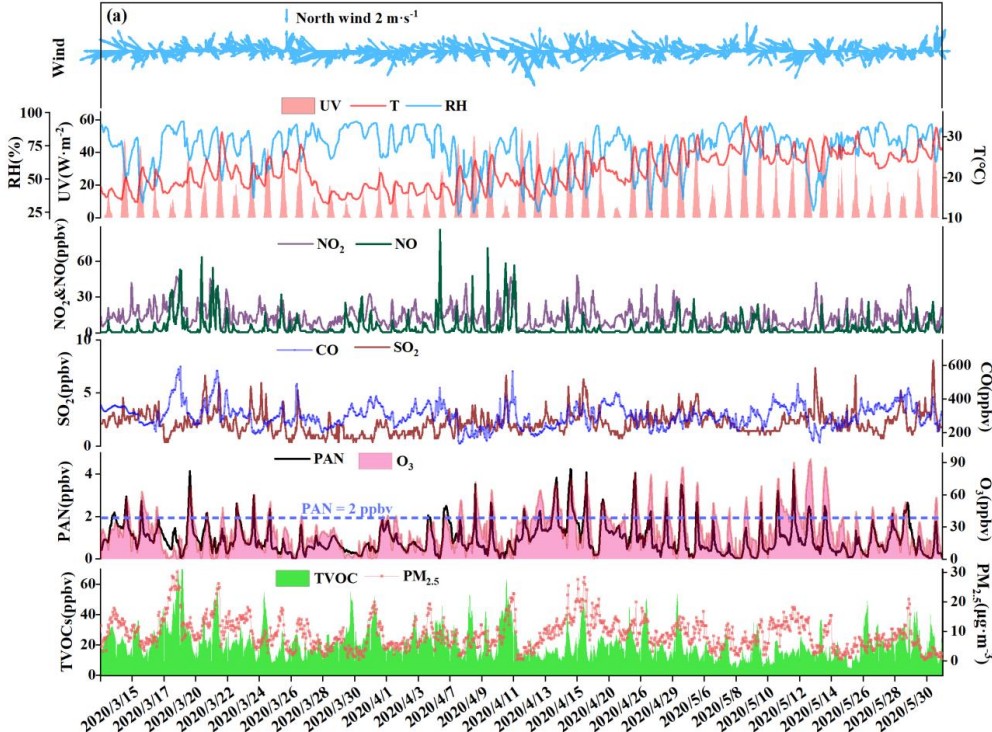


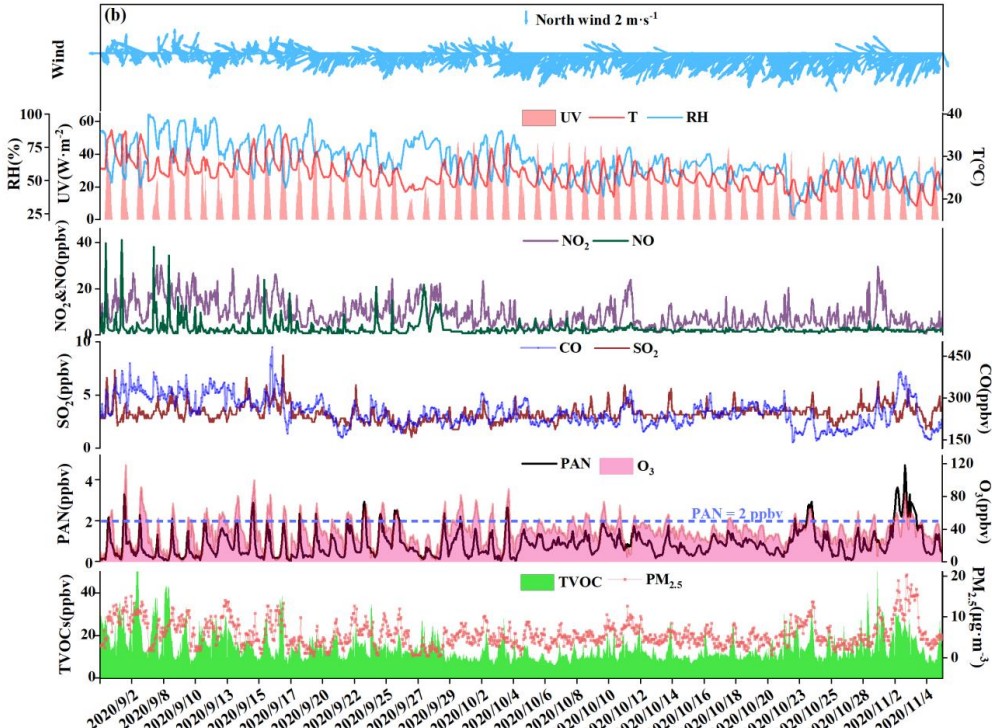

**Fig. 2. Time series of PAN, O₃, NOx, CO, SO₂, TVOCs, PM₂.₅, and meteorological parameters in (a) spring and (b) autumn.**

The time series of air pollutants and meteorological parameters are shown in Fig. 2. The average levels of PAN in autumn (0.87±0.66 ppbv) were comparable to that in spring (0.96±0.73 ppbv), while $O_3$ mixing ratios in autumn (37.22±16.89 ppbv) were 1.39 times higher than that in spring (26.73±18.63 ppbv). PAN and $O_3$ are produced by the photochemical reactions of VOCs and NOx, thus they usually show a relatively close relationship ($R^2 \geq 0.49$, Fig. S1). The PAN level (0.92±0.69 ppbv) in Xiamen was lower than that of megacities such as Beijing (3.79±3.26 ppbv) (Xu et al., 2021), Jinan (2.54 ppbv) (Liu et al. 2018), Santiago (6.4 ppbv) (Rubio et al., 2005) and Chongqing (2.05 ppbv) (Sun et al., 2020), and was comparable to the coastal cities with relatively clean air, including Shenzhen (1.01±0.94 ppbv) (Xia et al., 2021), and Qingdao (0.81 ppbv) (Liu et al., 2021).

The averaged values of PAN and NO, $NO_2$, CO, TVOCs in spring were 1.70, 1.32, 1.21, and 1.46 times higher than those in autumn, respectively. The details of measured VOCs were provided in Table S2. Alkanes, OVOCs, aromatics and halocarbons accounted for about 90% of total VOCs, suggesting the impacts of atmospheric oxidation capacity and marine emissions in coastal regions (Liu et al., 2020a; Liu



et al., 2020b). The wind directions in late spring and early autumn were messy due to the season switch.
The wind rose charts showed that the wind direction frequencies with relatively high wind speed in spring
and autumn were southeast wind and northeast wind (Fig. S2). Although the frequency of northwest wind
(NNW) also accounted for a certain proportion, the NNW speeds were generally slow, and the direction
of the NNW was mainly rural residential and mountainous areas with less anthropogenic emissions, so
that it was not the focus of this research. The ultraviolet radiation (UV), WS and T in spring (15.32 W·m$^{-2}$;
1.96 m·s$^{-1}$; 21.51 ℃) were weaker than those in autumn (18.43 W·m$^{-2}$; 3.01 m·s$^{-1}$; 25.85 ℃), and RH
and P in spring (73.25 %; 1010.71 hPa) were higher than that in autumn (65.21 %; 1008.71 hPa). These
meteorological conditions carried by the WPSH (high T, low RH, and stagnant weather conditions) were
conducive to the photochemical reaction and accumulation of air pollutants in autumn (Wu et al., 2019;
Xia et al., 2021). High mixing ratios of PAN precursors in spring were conducive to the continuous and
stable production of PAN, and the high air temperature in autumn accelerated the thermal decomposition
of PAN. However, the O$_3$ levels in autumn were higher than that in spring, attributing to the influence of
strong photochemical reaction conditions, regional transport from the Yangtze River Delta region or
increased atmospheric background levels (Monks, 2000). High O$_3$ values in both seasons were
concentrated on the wind direction of southeast and northeast (Fig. S3). We calculated the PAN lifetime
in our previous study (Hu et al., 2020). The PAN lifetimes were 6.39 and 2.02 hours in spring and autumn,
respectively. High PAN values easily happened in the wind direction of the southeast with low wind speed
(<3 m·s$^{-1}$) in spring, and northeast with a relatively high wind speed (from Quanzhou city, an industrial
city adjacent to Xiamen) in autumn. Accordingly, O$_3$ showed obvious characteristics of long-range
transport, and PAN pollution was mainly from local production/accumulation.

Based on the above analysis, we found that the photochemical reactions were still intense and even

stronger under the condition of low precursor mixing ratios. Although the precursor mixing ratios of PAN
and O$_3$ in spring were significantly higher than those in autumn (P<0.01), the PAN mixing ratios were
comparable to and O$_3$ mixing ratios were much higher in autumn than those in spring, respectively.
Therefore, it is very necessary to furtherly explore the key influencing factors and their formation
mechanisms.

### 3.2 The influencing factors of PAN using the GAM

PAN levels are not only related to chemical reactions in the boundary layer, but also affected by

meteorological conditions (Hu et al., 2020). According to the collinearity analysis (He and Lin, 2017), the



meteorological parameters (UV, T, RH and WS) and other air pollutants (NO, TVOCs, $PM_{2.5}$ and $O_x$)
were considered into the multiple-factor GAM model (Table S3). As shown in Table 1, the adjusted $R^2$
and deviance explained for the smoothed variables of the multiple-factor GAM model were 0.70 and 72%
in spring, 0.60 and 63% in autumn. According to the F-values, the orders of the explanatory variables in
spring and autumn were UV (60.64) > Ox (57.65) > T (17.55) > $PM_{2.5}$ (9.94) > TVOCs (9.52) > NO
(8.73)> WS (7.42) > RH (3.4) and Ox (58.45) > TVOCs (21.63) > T (20.46) > $PM_{2.5}$ (14.53) > RH (10.99) >
UV (7.13) > NO (4.16) >WS (2.55), respectively.

Response curves of the PAN mixing ratio to explanatory factors in the multiple-factor model were

presented (Fig. 3 and Fig. S4). In spring, except for UV and T, the degrees of freedom (df) of the
explanatory variables were greater than 1, indicating the non-linear relationships between explanatory
variables and PAN. The PAN in both seasons showed a downward trend with the increase of NO, while
the PAN in spring was unchanged with the fluctuation of NO. As we all know, the reaction of PA+NO is
one of the most important loss pathways of PA, suggesting the fact that NO consumed PAN indirectly
(Liu et al., 2021). Ox had a positive correlation with PAN, representing the promotion effects of
atmospheric oxidation capacity on PAN formation. With the increase of TVOCs and $PM_{2.5}$ levels, PAN
showed an upward trend. UV also had a significant positive correlation with PAN. However, the air
temperature had a significant negative correlation with PAN, due to the thermal decomposition of PAN.
In addition, when RH was more than 40%, the increase of RH was unfavorable for PAN production in
both seasons. Some studies also found that high water vapor content could remove PAN and its precursors,
thus reducing solar ultraviolet radiation to affect the photochemical reaction of PAN (Yan et al., 2018; Ma
et al., 2020). Simultaneously, RH could affect the heterogeneous reactions by influencing viscosity and
the phase diffusion of aerosol particulates (Li et al., 2013; Slade et al., 2014). Overall, the major factors
of PAN formation in spring were UV, Ox, and T, while those factors including Ox, TVOCs, T and $PM_{2.5}$
played important roles in autumn.

**Table 1 Estimated degree of freedom (Edf), degree of reference freedom (Ref. df), P-value, F-value, deviance**
**explained (%), adjusted $R^2$, deviance contribution (%) for the smoothed variables in the multiple-factor GAM**
**model.**

| Smoothed | Spring | | | | Autumn | | | |
|---|---|---|---|---|---|---|---|---|
| variables | Edf | Ref.df | F-value | P-value | Edf | Ref.df | F-value | P-value |
| NO (ppbv) | 5.21 | 6.26 | 8.73 | 0.00 | 1.11 | 1.21 | 4.16 | 0.03 |
| Ox (ppbv) | 4.73 | 5.85 | 57.65 | 0.00 | 4.84 | 5.98 | 58.45 | 0.00 |
| TVOCs (ppbv) | 7.14 | 8.19 | 9.52 | 0.00 | 4.08 | 5.06 | 21.63 | 0.00 |
| $PM_{2.5}$ (ppbv) | 5.73 | 6.86 | 9.94 | 0.00 | 1.53 | 1.90 | 14.53 | 0.00 |
| UV (W·m$^{-2}$) | 1.00 | 1.00 | 60.64 | 0.00 | 4.38 | 5.38 | 7.13 | 0.00 |

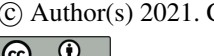


| | | | | | | | | |
|---|---|---|---|---|---|---|---|---|
| T (℃) | 1.00 | 1.00 | 17.55 | 0.00 | 2.73 | 3.46 | 20.46 | 0.00 |
| RH (%) | 6.78 | 7.87 | 3.40 | 0.00 | 6.56 | 7.68 | 10.99 | 0.00 |
| WS (m·s⁻¹) | 5.22 | 6.37 | 7.42 | 0.00 | 5.12 | 6.28 | 2.55 | 0.02 |

| Deviance explained (%)=80%; Adjust $R^2$=0.79 | Deviance explained (%)=72%; Adjust $R^2$=0.70 |
|---|---|


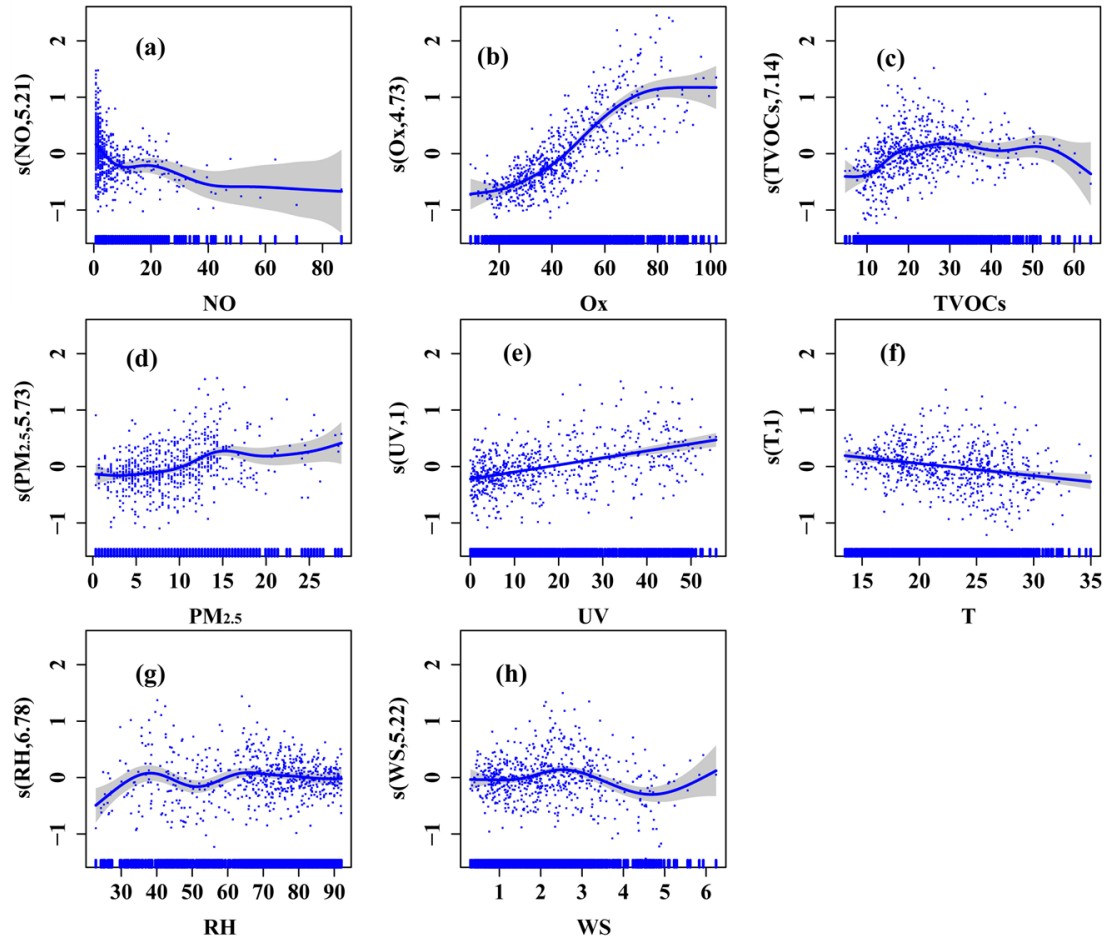


**Fig. 3. Response curves (spring) in the multiple-factor model of PAN mixing ratio to changes in (a) NO, (b) Ox**
**(Ox=O₃+NO₂), (c) TVOCs, (d) PM₂.₅, (e) ultraviolet radiation (UV), (f) air temperature (T), (g) relative humidity**
**(RH), and (h) wind speed (WS). The y-axis is the smoothing function values. For example, s(NO, df) shows the**
**trend in PAN when NO changes, and the number of df is the degree of freedom. The x-axis is the influencing factor,**
**and the shaded area around the solid red line indicates the 95% confidence interval of PAN. The blue vertical**
**short lines represent the concentration distribution characteristics of the explanatory variables (units: NO (ppbv),**
**Ox (O₃+NO₂) (ppbv), TVOCs (ppbv), PM₂.₅ (ppbv), UV (W·m⁻²), T (℃), RH (%), WS (m·s⁻¹)).**

**3.3. Formation mechanism of PAN**
**3.3.1 Diurnal variation during episodes and non-episodes**



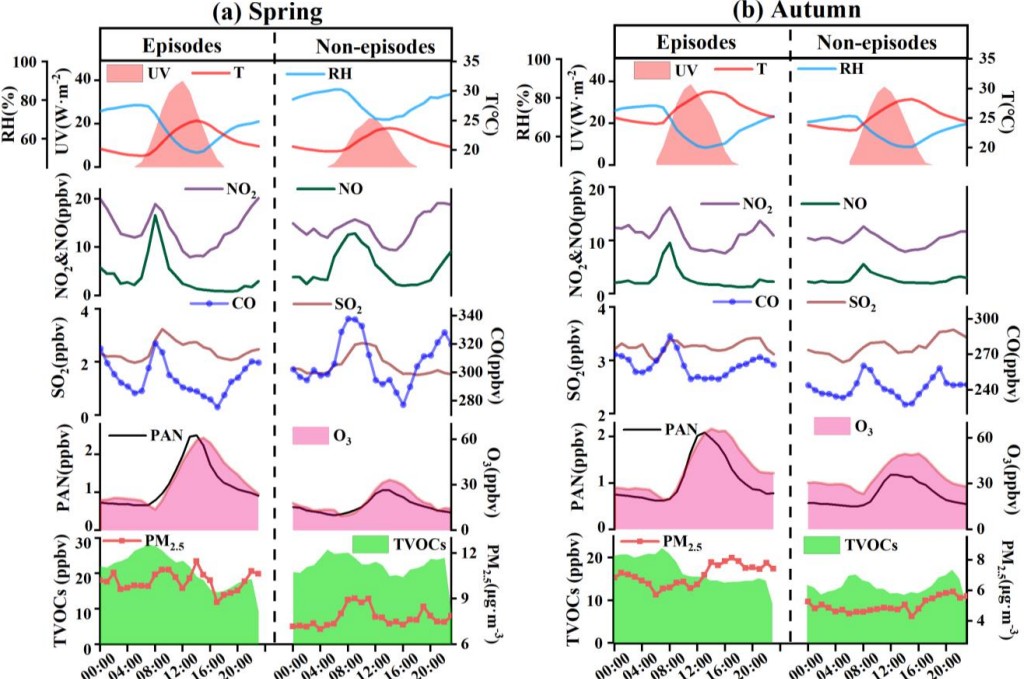

**Fig. 4. Diurnal trends of PAN, O$_3$, TVOCs, PM$_{2.5}$, other trace gases and meteorological parameters during**
**episodes and non-episodes in (a) spring and (b) autumn, respectively.**

Throughout the 53-days campaign, 30 and 21 days (i.e., 57% and 40%) with the peak values of PAN
exceeding 2 ppbv were observed in spring and autumn, respectively. The scenarios of episodes and non-
episodes were classified, according to the previous method (Xue et al., 2014). Diurnal variations of air
pollutants and meteorological parameters during episodes and non-episodes are shown in Fig. 4. PAN
reached a maximum value at 14:00 (episodes) and 13:00 (non-episodes) in spring, while 13:00 (episodes)
and 12:00 (non-episodes) were observed in autumn, related to UV radiation intensity. Compared with
PAN, the O$_3$ peak was delayed by 1-2 hours, due to the accelerated decomposition of PAN at high air
temperatures. The lowest PAN and O$_3$ mixing ratios in the early morning were caused by the all-night
thermal decomposition without the photochemical formation and the NO titration during rush hour traffic,
respectively (Zhang et al., 2015a; Elshorbany et al., 2008). A relatively broad peak of PAN and O$_3$ in
autumn reflected the influence of regional transport (Zeng et al., 2019). Due to the photochemical
reactions, the precursors of CO, NOx and VOCs were consumed during the daytime, and were
accumulated during the nighttime with weak solar radiation. CO, NOx and TVOCs showed the highest
levels at around 08:00 LT due to the nighttime accumulation and vehicle exhaust, but showed relatively
low levels during the daytime, emphasizing the importance of frequent human activities and weather





conditions. PAN and O$_3$ are secondary pollutants depending on both the levels of precursors and
photochemical reactions. Therefore, the trends of PAN and O$_3$ were opposite to that of their precursors.

In autumn, averaged PAN and O$_3$ during episodes (PAN: 1.08±0.87 ppbv, and O$_3$: 40.06±20.27 ppbv)

were higher than those during non-episodes (PAN: 0.74± 0.41 ppbv, and O$_3$: 35.36±13.95 ppbv).
Meanwhile, some air pollutants and meteorological parameters during episodes were 1.03-1.40 times
higher than those during non-episodes. The rainfall in Xiamen is more frequent in spring (Hu et al., 2020),
leading to the obvious differences in UV and RH levels between episodes and non-episodes. In spring,
the precursors (CO, NOx, TVOCs) of PAN during episodes were 1.04-1.49 times lower than those during
non-episodes. Moreover, the PAN and O$_3$ mixing ratios during episodes (PAN: 1.20±0.81 ppbv, and O$_3$:
32.92±19.81 ppbv) were still significantly higher than those during non-episodes (PAN: 0.64±0.43 ppbv,
and O$_3$: 18.65±13.16 ppbv), attributing to the favorable meteorological conditions of photochemical
reactions (strong UV, high T and low RH). These results further explained that UV, Ox, and T in spring
and Ox, TVOCs, T and PM$_{2.5}$ in autumn played important roles in the formation of PAN based on the
GAM analysis.

**3.3.2. Formation and loss of PA radical**

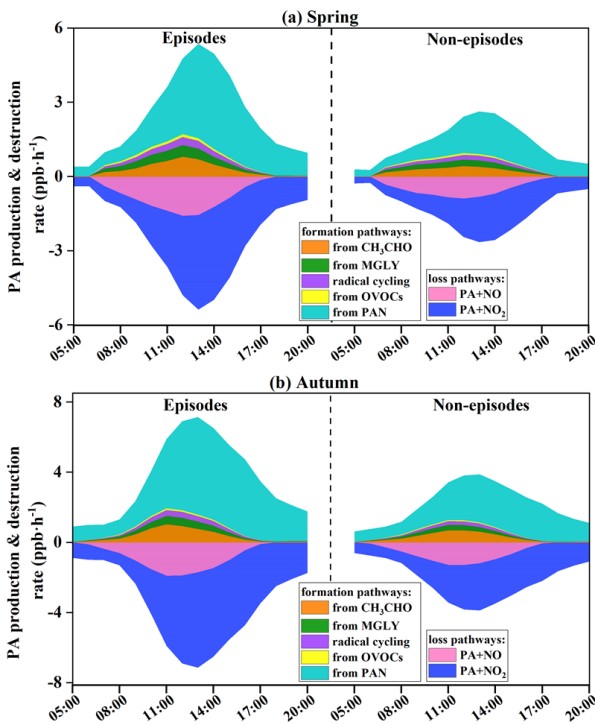


**Fig. 5. Formation and destruction rates of PA radical (hence PAN) during episodes and non-episodes in (a) spring**





**and (b) autumn, respectively.**

The formation and sink pathways of PA radical were further explored under different pollution
scenarios (Fig. 5). Both the PA (hence PAN) production and destruction rates during episodes were 1.80
times higher than those during non-episodes. Combined with the analysis of Section 3.3.1, PA production
rates during the daytime (06:00-17:00 LT) in autumn were 1.58 times higher than that in spring, even
though the precursor levels in autumn were much low compared to those in spring. These results indicated
favorable meteorological conditions producing and accumulating pollutants and local transport bringing
high air pollutants from Quanzhou city or urban plumes were dominant factors for the strong PAN
production rates. The thermal decomposition of PAN to PA radical in autumn accounted for 77±12%
(episodes) and 73±16% (non-episodes) of total PA production, as well as 70±12% (episodes) and 64±15%
(non-episodes) in spring, attributing to the relatively high air temperature and UV intensity. The thermal
decomposition of PAN peaked at around 13:00~14:00 LT, when the air temperature was the highest in the
day, and the pathways without considering the transform between PA and PAN peaked at noontime around
12:00 LT, when the solar radiation was the highest and photochemical reactions became the most intensive.
The average daytime PAN production rate from $CH_3CHO$ by reacting with OH and $NO_3$ contributed
0.36±0.25 ppb $h^{-1}$ and 0.24±0.13 ppb $h^{-1}$ during episodes and non-episodes in spring. While the rate of
0.46±0.35 ppb $h^{-1}$ and 0.34±0.24 ppb $h^{-1}$ during episodes and non-episodes were observed in autumn.
The second production reaction was photolysis and oxidation by OH and $NO_3$ of MGLY (episode:
0.25±0.15 ppb $h^{-1}$ and non-episodes: 0.17±0.08 ppb $h^{-1}$ in spring; episode: 0.24±0.17 ppb $h^{-1}$ and non-
episodes: 0.16±0.11 ppb $h^{-1}$ in autumn). Then, the processes of radical cycling including RO radical
decomposition and reactions of acyl peroxy radicals with NO were also the important sources to produce
PA, with the contributions of 20±3% and 18±3% in spring and autumn. PA from the other OVOCs (not
including $CH_3CHO$, MGLY, MVK, MACR and acetone) through reactions of photolysis and oxidation
by OH, $NO_3$, and $O_3$, accounted for 7±2% and 6±1% in spring and autumn, respectively. Other reactions
of acetone, MVK, MACR, MPAN and isoprene had a minor contribution (around 1% in total) to PA
formation. In contrast, the major contributor of PAN destruction rate was PA+$NO_2$ (69±16% in spring and
73±14% in autumn), followed by PA+NO (31±17% and 27±13%), while the other reactions with $NO_3$,
$HO_2$ and $RO_2$ contributed limitedly (around 0.1% of the total).

**3.3.3. Sensitivity of PAN precursors**

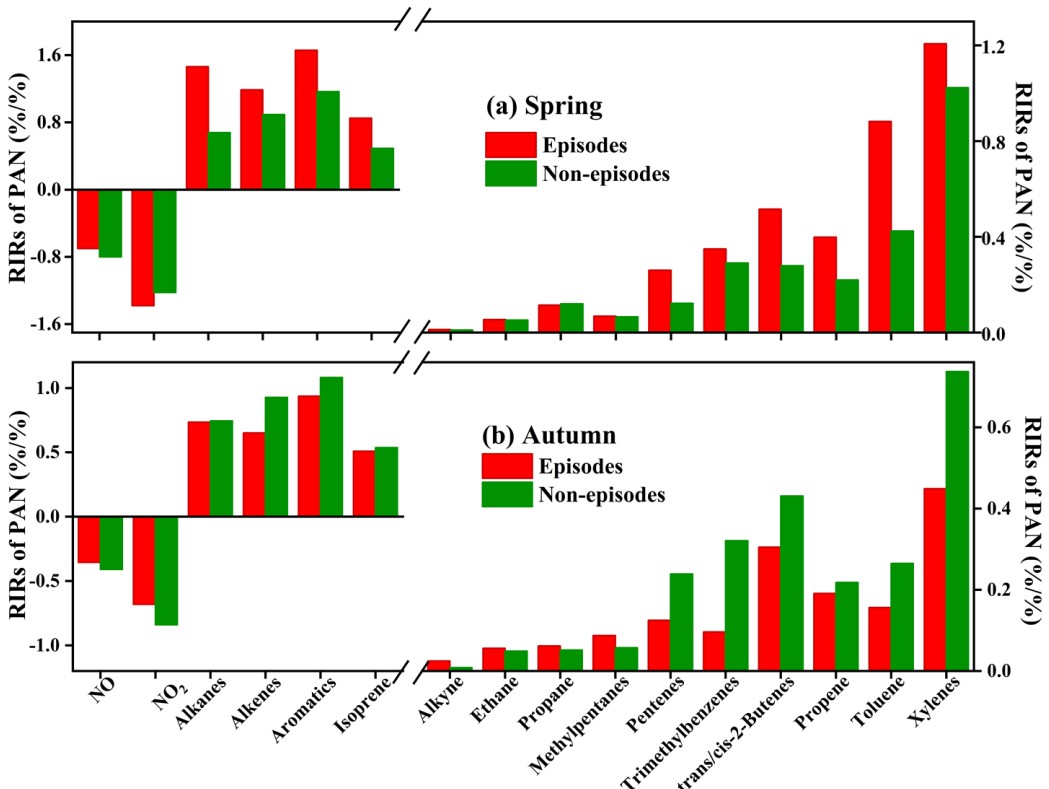

**Fig. 6. The OBM-MCM calculated relative incremental reactivity (RIR) for major PAN precursor groups and**
**specific species in (a) spring and (b) autumn during the daytime (06:00-17:00 LT).**

The OBM-MCM model analysis could be used to examine the relationship between PAN and its
precursors, and quantify the contribution of first-generation precursors (Liu et al., 2021; Cardelino and
Chameides, 1995). The relative incremental reactivities (RIRs) for $O_3$ and PAN are shown in Fig. 6 and
Fig. S5. The PAN production was highly VOCs-sensitive, while the RIRs of NO and $NO_2$ were negative
ranging from −0.17 to −1.94%/% during the daytime (06:00-17:00 LT). This consisted of the fact that
high dense mobiles resulted in the large emissions of vehicle exhausts in Xiamen city. The ratio of
VOCs/NOx (1.11±0.32) also convinced NOx was not the limited factor on the photochemical reaction
(Tan et al., 2019). In suburban or rural areas, the transition regime and NOx-sensitive for PAN and $O_3$
production were usually found (Xue et al., 2014; Liu et al., 2021). Zeng et al. (2019) found $NO_2$-positive
and NO-negative to PAN formation in a suburban of Hong Kong, consisting with the fact that $NO_2$ directly
produced PAN and NO consumed PA radical inhibiting PAN formation.
As shown in Fig.6, aromatics showed the largest RIRs for PAN in spring (1.41%/%) and autumn
(1.03%/%), followed by alkanes (1.04%/% in spring and 0.78%/% in autumn), Alkenes (1.04%/% and





0.74%/%), and isoprene (0.67%/% and 0.52%/%). The sensitivities of PAN precursors in spring were
1.37-2.07 times higher than those in autumn, due to the large percentages of PAN decomposition at high
air temperatures in autumn. In spring, the weak solar radiation led to poor photochemical reactions, so
the RIRs of PAN during non-episodes were lower than that during episodes. However, the PAN
sensitivities during episodes were lower than those during non-episodes, attributed to the rapid PAN
decomposition in autumn (Liu et al., 2021). In addition, RIRs of VOCs and NOx for PAN were
significantly higher than that of $O_3$ (Fig. S5). For RIRs of VOCs, except for air temperature, the different
formation mechanisms of PAN and $O_3$ should be considered. Only a small part of the VOCs could produce
PA to form PAN, thereby, the VOCs were insufficient to produce PAN (Fischer et al., 2014). For RIRs of
NOx, $O_3$ was produced from the $NO_2$ conversion process, and was also rapidly consumed by NO titration.
High mixing ratios of VOCs and NOx enhanced the PAN formation, even though a pathway of NO
destructed PAN, which was negligible compared to thermal decomposition. For this reason, the RIRs of
NOx for PAN were higher than those for $O_3$.

In addition, the key first-generation VOCs species (including xylenes, toluene trans/cis-2-butenes,

trimethylbenzenes, propene, pentenes, and methypentanes) governing PAN production were further
identified (Fig. 6). The results suggested that the reduction of aromatics, alkenes and alkanes with ≤5
carbons could effectively decrease PAN pollution.

**3.4. Impacts of PAN on $O_3$ formation**



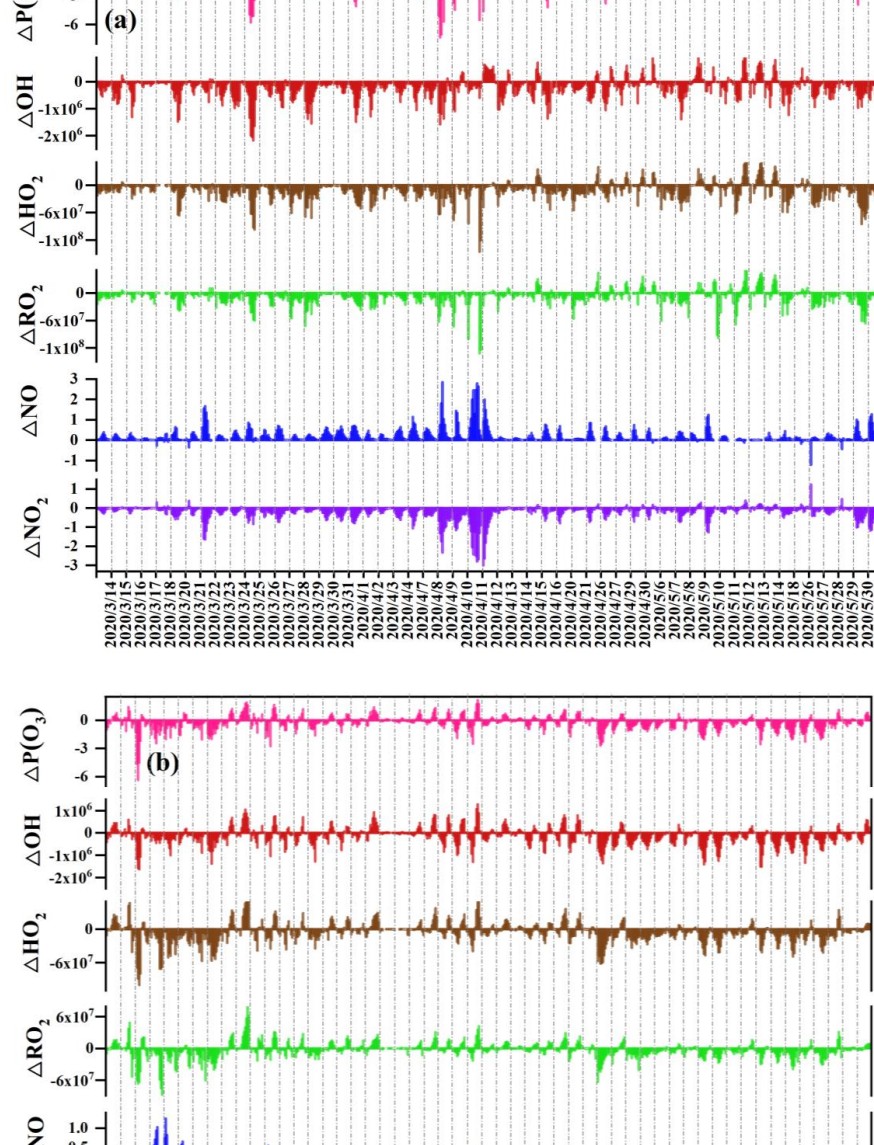



**Fig. 7. The differences of O₃ net production △P(O₃), △OH, △HO₂, △RO₂, △NO and △NO₂ between the SC1**


**and the SC2 during the daytime (06:00-17:00) in (a) spring and (b) autumn (Unit: ppbv·h⁻¹ for △P(O₃); ppbv for**


**△NO and △NO₂; molecules·cm⁻³ for △OH, △HO₂ and △RO₂). The SC1 scenario was the base scenario putting**




**all detected data (i.e. VOCs, trace gases and meteorological parameters) into the model with all reaction pathways**
**of the MCM mechanism, and the SC2 disabled the PAN chemistry, which is the only difference between SC1 and**
**SC2.**

PAN could affect $O_3$ production by acting as a temporary source of NOx or sink of PA radical to

affect precursors and radical chemistry in the troposphere (Xia et al., 2021). To quantify the changes of
$O_3$ in response to PAN chemistry in the coastal city, two parallel scenarios (SC1 and SC2) were conducted
based on the OBM model. The SC1 was the base scenario putting all detected data (i.e. VOCs, trace gases
and meteorological parameters) into the model with all reaction pathways (as the description in Section
2.2), and the SC2 disabled the PAN chemistry, which is the only difference between SC2 and SC1. Figure
7 shows the differences of $O_3$ net production rates $\triangle P(O_3)$, $\triangle OH$, $\triangle HO_2$, $\triangle RO_2$, $\triangle NO$ and $\triangle NO_2$
between the SC1 and the SC2. Negative and positive values represented the inhibition and promotion
effects of PAN photochemistry on $O_3$ formation, respectively. Overall, PAN mostly inhibited the $O_3$
formation during the observation days. $\triangle P(O_3)$ had significantly positive correlations with $\triangle OH$
($R^2$=0.96 in spring and 0.95 in autumn), $\triangle HO_2$ ($R^2$=0.91 and 0.96), $\triangle RO_2$ ($R^2$=0.86 and 0.86) and $\triangle NO_2$
($R^2$=0.72 and 0.85), and negative correlation with $\triangle NO$ ($R^2$=-0.63 and -0.65). As shown in Fig. S6, the
promotion effects of PAN on $O_3$ mainly happened during the periods of 11:00-16:00 LT, and most of them
concentrated on PAN pollution episodes. The percentage of negative $\triangle P(O_3)$ values were 83% and 69%
in spring and autumn, defined as "inhibition effect stages". While the positive $\triangle P(O_3)$ values accounted
for 17% and 31% in spring and autumn, defined as "promotion effect stages".

Figure 8 shows the variations of modeled $P(O_3)$, $O_3$ budgets and ROx on the inhibition and promotion

effect stages in spring and autumn. The abundance of ROx in autumn ($2.85\times10^8$ molecules $cm^{-3}$) was
higher than that in spring ($2.08\times10^8$ molecules $cm^{-3}$) during inhibition effect stages, while the $P(O_3)$ value
in autumn (5.24 ppbv $h^{-1}$) was higher than that in spring (4.88 ppbv $h^{-1}$). On the contrary, the level of ROx
in spring ($4.81\times10^8$ molecules $cm^{-3}$) was higher than that in autumn ($4.20\times10^8$ molecules $cm^{-3}$) during
promotion effect stages, and the $P(O_3)$ value (5.95 ppbv $h^{-1}$) in spring was higher than that in autumn
(5.76 ppbv $h^{-1}$). The results indicated that high ROx concentration was an important factor for the
formation of $O_3$. In the case of closing PAN photochemistry, the $P(O_3)$ increased 1.20 and 1.12 times
during inhibition effect stages and decreased 1.09 and 1.08 times during promotion effect stages in spring
and autumn, respectively (Fig. 8a). This was consistent with the corresponding changes of ROx radical
(Fig. 8b). During the inhibition effect stages, the averaged concentrations of OH, $HO_2$ and $RO_2$ increased
1.05, 1.16, and 1.17 times in spring, and increased 1.04, 1.10, and 1.12 times in autumn. During the
promotion effect stages, the averaged concentrations of OH, $HO_2$ and $RO_2$ decreased 1.02, 1.03, and 1.06





times in spring, and decreased 1.02, 1.04, and 1.05 times in autumn. These results indicated that the
changes in ROx dominated the $P(O_3)$ trend without PAN photochemistry. Furthermore, the $P(O_3)$ level
during promotion effect stages (5.95 ppbv h$^{-1}$ in spring, 5.76 ppbv h$^{-1}$ in autumn) was higher than that
during inhibition effect stages s (4.88 ppbv h$^{-1}$ in spring, 5.24 ppbv h$^{-1}$ in autumn). For model-simulated
$P(O_3)$ and $O_3$ budgets (Fig. 8a), $HO_2+NO$ (account for 70±4%) and $RO_2+NO$ (30±6%) were the main
pathways of $O_3$ formation, and the main loss reactions were $OH+NO_2$ (83±12%).

PAN competed with $O_3$ precursors and terminated the radical chain to suppress $O_3$ formation by

decreasing the ROx production during the inhibition effect stages. During the promotion effect stages, the
intensive atmospheric oxidation capacity and photochemical reaction enhance the ROx formation rates
from PAN to promote $O_3$ formation (Fig. 8b).

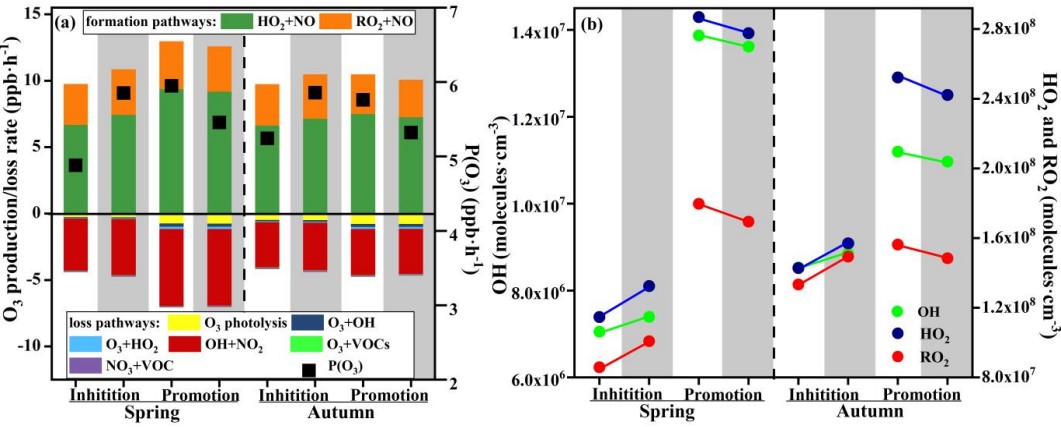


**Fig. 8. Model-simulated (a) net $O_3$ production rate and $O_3$ budgets, (b) OH, $HO_2$ and $RO_2$ on the inhibition effect**
**stages and promotion effect stages. Note: the white background parts represent the SC1 scenarios using the MCM**
**mechanism, and the gray background parts represent the SC2 scenarios using the MCM mechanism with PAN**
**chemistry disabled.**

Table S4 showed the air pollutants and meteorological parameters during the inhibition effect stages

and promotion effect stages. In detail, the mixing ratios of CO and the precursors of $O_3$ and PAN during
the inhibition effect stages were significantly higher than those during the promotion effect stages.
However, the $PM_{2.5}$ level during the inhibition effect stages was relatively lower than that during the
promotion effect stages, reflecting the influence of heterogeneous reactions on $PM_{2.5}$ by suppling key
photochemical oxidants to enhance PAN production (Xu et al., 2021). In addition, $SO_2$ and wind speed
were comparable during the two scenarios. During the promotion effect stages, UV and T were
significantly high, while P and RH were significantly low (P<0.01). Meanwhile, the PAN (1.89 in spring,
1.58 ppbv in autumn) and $O_3$ (50.26 ppbv in spring and 53.51 in autumn) under the promotion effects



were higher than those under the inhibition effects (PAN: 1.04 and 0.84 ppbv; $O_3$: 27.32 and 36.42 ppbv
in spring and autumn, respectively).
In general, ROx radicals dominated the atmospheric oxidative capacity and were the indicators of
atmospheric photochemical reaction (Li et al., 2018). According to Section 3.2 of GAM analysis, we
chose the factors of NO, TVOCs, $PM_{2.5}$, UV, T, RH, WS, and $\triangle ROx$ ($\triangle ROx = \triangle OH + \triangle HO_2 + \triangle RO_2$), to
discuss the key influencing factor under promotion effect stages. Here, the $\triangle P(O_3)$ rate and the relevant
influencing factors were set as the response and explanatory variables, respectively. Table 2 showed the
influencing factors on $\triangle P(O_3)$ under promotion effects in spring and autumn. The factors that did not pass
the significance test were deleted. As the adjusted model showed, the adjusted $R^2$ and deviance explained
for the smoothed variables in four GAM models ranged from 0.67~0.78 and 70%~80%, verifying the
good fitting effect of the multiple-factor GAM model. According to the F-values, the effects of $\triangle ROx$
(21.56 in spring; 45.45 in autumn) and UV (9.66 in spring; 30.55 in autumn) were the main factors leading
to the promotion effect in both seasons. Both $\triangle ROx$ and UV had significant positive non-linear
relationships with $\triangle P(O_3)$ during promotion effect stages in both seasons (Fig. S7 and S8). The minor
influences of WS and T were observed in autumn. The promotion effects easily happened during periods
of favorable meteorological conditions for photochemical reactions.
Liu et al. (2021) found that PAN photochemistry inhibited $O_3$ production under low-NOx and low-
ROx conditions, and promoted $O_3$ formation under high-NOx. However, in this study, sufficient NOx
would not be the limited factor and the change of NOx could be ignored. Whether PAN photochemistry
suppressed or enhanced $O_3$ production mainly depended on the meteorological conditions of
photochemical reaction and the ROx levels.

**Table 2 Estimated degree (during promotion effect scenarios in spring and autumn) of freedom (Edf), degree of**
**reference (Ref. df), P-value, F-value, deviance explained (%), adjusted $R^2$, deviance contribution (%) for the**
**smoothed variables (including NO, $\triangle ROx$, TVOCs, $PM_{2.5}$, UV, T, RH, and WS) in the multiple-factor GAM model.**

| Smoothed variables | Incipient | | | | Adjusted | | | |
|---|---|---|---|---|---|---|---|---|
| | Edf | Ref.df | F-value | P-value | Edf | Ref.df | F-value | P-value |
| **Promotion effect stages in spring** | | | | | | | | |
| NO (ppbv) | 5.58 | 6.39 | 2.09 | 0.06 | | Delete | | |
| ROx (molecules·cm$^{-3}$) | 5.99 | 7.06 | 22.88 | 0.00 | 5.72 | 6.83 | 21.56 | 0.00 |
| TVOCs (ppbv) | 1.14 | 1.26 | 0.60 | 0.40 | | Delete | | |
| $PM_{2.5}$ (ppbv) | 1.98 | 2.51 | 2.62 | 0.07 | | Delete | | |
| UV (W·m$^{-2}$) | 3.89 | 4.80 | 7.40 | 0.00 | 2.98 | 3.73 | 9.66 | 0.00 |
| T (℃) | 1.00 | 1.00 | 1.88 | 0.17 | | Delete | | |
| RH (%) | 1.00 | 1.00 | 0.86 | 0.36 | | Delete | | |
| WS (m·s$^{-1}$) | 1.41 | 1.71 | 3.03 | 0.13 | | Delete | | |



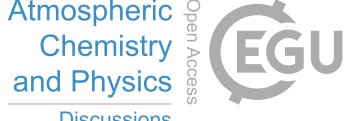

| Promotion effect stages in autumn | | | | | | | | |
|---|---|---|---|---|---|---|---|---|
| NO (ppbv) | 1.15 | 1.28 | 0.20 | 0.66 | Delete | | | |
| ROx (molecules·cm$^{-3}$) | 7.10 | 8.06 | 41.04 | 0.00 | 7.37 | 8.26 | 45.45 | 0.00 |
| TVOCs (ppbv) | 1.00 | 1.00 | 0.00 | 0.97 | Delete | | | |
| PM$_{2.5}$ (ppbv) | 1.00 | 1.00 | 0.53 | 0.47 | Delete | | | |
| UV (W·m$^{-2}$) | 3.11 | 3.87 | 28.90 | 0.00 | 3.07 | 3.83 | 30.55 | 0.00 |
| T (℃) | 2.26 | 2.87 | 4.73 | 0.01 | 2.28 | 2.88 | 7.41 | 0.00 |
| RH (%) | 1.50 | 1.87 | 0.58 | 0.62 | Delete | | | |
| WS (m·s$^{-1}$) | 4.67 | 5.76 | 2.73 | 0.02 | 4.53 | 5.60 | 3.66 | 0.00 |

## 4. Conclusions

Field observation was continuously conducted in spring and autumn in a coastal city of Southeast China. We clarified the seasonal variations of PAN pollution, formation mechanisms, influencing factors and impacts on O$_3$ production. The average levels of PAN in autumn were lower than that in spring, while the O$_3$ showed the opposite characteristics. The multiple-factor GAM model showed that the key factors on PAN mixing ratio were UV, Ox, and T in spring, while Ox, TVOCs, T and PM$_{2.5}$ played important roles in PAN formation in autumn. The MCM model is an ideal tool to explore PAN photochemical formation and its key precursors at the species level and provides more relevant suggestions for reducing photochemical pollution. The controlling emissions of aromatics and alkenes with ≤5 carbons were benefit for PAN pollution mitigation, and carbonyl compounds especially acetaldehyde were dominant in the PAN production mechanism. PAN presented the inhibition or promotion effects on O$_3$ under different environmental conditions. The promotion effects of PAN on O$_3$ mainly happened during the periods of 11:00-16:00 LT, most of which concentrated on PAN pollution episodes. According to the GAM analysis, the levels of ROx and UV were the main factors leading to the promotion effects in both seasons. Overall, PAN stimulated O$_3$ formation under high levels of UV, T and ROx in the coastal city. These results indicate that the monitoring of PAN and its precursors and the quantification of its impacts on O$_3$ formation have significant guidance on photochemical pollution control. The scientific analysis methods used in this study provide a reference for the research on the formation mechanism of PAN and O$_3$ in other regions.

**Authorship Contribution Statement**

Taotao Liu performed chemical modeling analyses of OBM-MCM and wrote the paper. Taotao Liu collected the data, contributed to the data analysis. Jinsheng Chen and Youwei Hong designed and revised the manuscript. Jinsheng Chen supported funding of observation and research. Gaojie Chen, Lingling Xu, Mengren Li, Yanting Chen, Xiaoting Ji, Chen Yang, and Yuping Chen contributed to discussions of results. Weiguo Huang, Quanjia Huang and Hong Wang provided part of the data in Xiamen.



**Acknowledgment:**

This study was funded by the Cultivating Project of Strategic Priority Research Program of the Chinese Academy of Sciences (XDPB1903), the FJIRSM&IUE Joint Research Fund (RHZX-2019-006), the Center for Excellence in Regional Atmospheric Environment, CAS (E0L1B20201), the Xiamen Youth Innovation Fund Project (3502Z20206094), the foreign cooperation project of Fujian Province (2020I0038) and Xiamen Atmospheric Environment Observation and Research Station of Fujian Province.

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
