# Peer review of "Seasonal characteristics of atmospheric peroxyacetyl nitrate (PAN) in a coastal city"

_Atmospheric Chemistry and Physics, 2021_

## Author Comment (AC2)

**Response to Reviewers**

Comment on acp-2021-948

**RC2 Anonymous Referee #2**

The paper entitled "Seasonal characteristics of atmospheric peroxyacetyl nitrate (PAN) 1 in a coastal city 2 of Southeast China: Explanatory factors and photochemical effects" is generally well written, interesting and within the scope of ACP. However, the authors need to address a few minor comments, before the paper is suitable to be published in ACP.

**Response:** Thanks for your valuable comments. We have corrected this manuscript according to your suggestions.

Major comments:

I find the description of the VOC measurements a bit brief. Which classes of VOCs did the authors monitor with their GC and using which columns? It is not clear whether aldehydes and ketones were measured and if so which ones. Figure S5 doesn't discuss OVOCs as ozone precursors. While table S2 includes some OVOCs, it doesn't mention some important compounds like acetaldehyde, MVK, methylglyoxal and methacrolein. Where these compounds measured and if yes how? It appears to me that the authors are assuming that these are primarily secondary compounds that are derived from other primary emissions ethane, propene, isoprene and aromatics (line 57). If they are just model calculated and not measured then this could bias the results. While these compounds, no doubt have photochemical sources, there can also be substantial primary emissions particularly from various types of biomass burning (Pallavi et al. https://doi.org/10.5194/acp-19-15467-2019), industrial sources and, in the case of acetaldehyde also vegetation (Sarkar et al. 2017 https://doi.org/10.5194/acp-17-8129-2017)

**Response:** Thank you for your suggestions. We have added more details of the description of the VOC measurements in the manuscript, as follows. Meanwhile, the detailed compounds of 106 VOCs were shown in Table R1.

"A gas chromatography-mass spectrometer (GC-FID/MS, TH-300B, Wuhan, CN) was used for monitoring the atmospheric VOCs with a 1-hour time resolution. The instrument conducted sampling with a 30 L/min sampling rate, then samples were pre-concentrated by cooling to -160 °C in a cryogenic trap followed by heating to 100 °C, and subsequently transferred to the secondary trap by high-purity helium (He). The flame ionization detector (FID) detected the low-carbon (C2-C5) hydrocarbons by a PLOT (Al$_2$O$_3$/KCl) column (15 m × 0.32 mm × 6.0 μm); the other compounds were quantified using a DB-624 column (60 m × 0.25 mm × 1.4 μm). The instrument system can quantitatively analyze 106 VOCs in the ambient atmosphere, including 29 alkanes, 11 alkenes, one alkyne, 17 aromatics, 35 halogenated hydrocarbons, and 13 OVOCs. The single-point calibration was performed every day at 23:00 with the standard mixtures of PAMS and TO15, and multi-point calibration was performed one month. The detection limits of the measured VOCs were in the range of 0.02 ppbv to 0.30 ppbv, and the measurement precision was ≤10%".

Because the states of calibration gases of aldehydes and ketones are very unstable, thereby it is difficult to

offer calibration gases during observation campaigns in our observation sites. Hence, the nine compounds can not be quantified (Acetaldehyde, Propanal, Crotonaldehyde, Methacrolein, n-butanal, Benzaldehyde, Valeraldehyde, m-Tolualdehyde, Hexanal). However, this phenomenon is common in many studies (Huang et al., 2021; Yang et al., 2019; Liu et al., 2021). Hence, the OVOCs that you mentioned can not be measured. All compounds of 106 VOCs detected by the gas chromatography-mass spectrometer are shown in Table R1. Meanwhile, the observation-based model has high requirements on the quality the monitor data, and quantities of the pollutants inputting into the OBM model are limited, so VOCs with concentrations below the detection line were removed. Therefore, the VOCs in Table S2 were the all VOCs compounds that we used in the OBM model in our study.

The RIR values in Figure S5 were calculated by the observation data. As said above, some important aldehydes and ketones can not be considered, so the RIRs of OVOCs were underestimated. Hence, the discussion of RIRs of OVOCs is not included. In many relevant studies, their results of RIR also did not consider the OVOCs (Liu et al., 2021; Chen et al., 2020). In the future, we will calculate the RIR of OVOCs, because of our upgraded monitoring station with the OVOCs measurements.

Thank you for your suggestions. We agree with your opinions that the precursors of PAN of OVOCs have both primary and secondary sources (Sinha et al., 2019; Sarkar et al., 2017). When these OVOCs concentrations were not observed, the concentrations could be locally and reasonably calculated by the model. The three modules (initial value settings, observation data input, pre-ran for 2 days before running the OBM model) could constrain the unmeasured compounds reaching a steady-state and make the model localization. Some studies exploring the PAN formation mechanism based on OBM model also did not observe these OVOCs data (Chen et al., 2020, Liu et al., 2021; Xue et al., 2014). Meanwhile, we strongly agree with your idea and realized the importance of OVOCs, and some OVOCs of model calculation without measuring could bias the results. Hence, our team improved the monitoring of OVOCs in October 2021. A more optimized and complete monitoring system is also the future optimization goal of our model. Anymore, the index of agreement (IOA) judging the reliability of the model simulation results showed the performance of the OBM-MCM model was reasonably acceptable, and its detailed calculation is shown in the next question.

**Table R1 The detailed compounds of 106 VOCs and the detectors.**

| Compounds | Detector | Compounds | Detector |
|---|---|---|---|
| ethene | FID | p-ethyltoluene | MS |
| ethane | FID | 1,2,4-trimethylbenzene | MS |
| propane | FID | 1,2,3-trimethylbenzene | MS |
| iso-butane | FID | m-diethylbenzene | MS |
| n-butane | FID | p-diethylbenzene | MS |
| iso-pentane | FID | naphthalene | MS |
| n-pentane | FID | dichlorodifluoromethane | MS |
| cyclopentane | FID | chloromethane | MS |
| propene | FID | 1,1,2,2-tetrachloro-Ethane | MS |
| 1-butene | FID | vinylchloride | MS |
| cis-2-butene | FID | bromomethane | MS |
| trans-2-butene | FID | chloroethane | MS |
| 1-pentene | FID | trichlorofluoromethane | MS |
| trans-2-pentene | FID | 1,1,2-trichloro-1,2,2-trifluoroethane | MS |
| cis-2-pentene | FID | carbondisulfide | MS |
| acetylene | FID | dichloromethane | MS |
| isoprene | FID | cis-1,2-dichloroethene | MS |

| | | | |
|---|---|---|---|
| 2,2-dimethylbutane | MS | 1,1-dichloroethane | MS |
| 2,3-dimethylbutane | MS | trans-1,2-dichloroethene | MS |
| 2-methylpentane | MS | trichloromethane | MS |
| 3-methylpentane | MS | 1,1,1-trichloroethane | MS |
| n-hexane | MS | 1,2-dichloroethane | MS |
| 2,4-dimethylpentane | MS | carbon tetrachloride | MS |
| methylcyclopentane | MS | trichloroethene | MS |
| cyclohexane | MS | 1,2-dichloropropane | MS |
| 2-methylhexane | MS | dichlorobromomethane | MS |
| 2,3-dimethylpentane | MS | cis-1,3-dichloropropene | MS |
| 3-methylhexane | MS | trans-1,3-dichloropropene | MS |
| 2,2,4-trimethylpentane | MS | 1,1,2-trichloroethane | MS |
| n-heptane | MS | dibromochloromethane | MS |
| methylcyclohexane | MS | tetrachloroethene | MS |
| 2,3,4-trimethylpentane | MS | 1,2-ethylenedibromide | MS |
| 2-methylheptane | MS | chlorobenzene | MS |
| 3-methylheptane | MS | tribromomethane | MS |
| n-octane | MS | Ethane, 1,1,2,2-tetrachloro- | MS |
| n-nonane | MS | 1,3-dichlorobenzene | MS |
| n-decane | MS | chlorotoluene | MS |
| n-undecane | MS | 1,4-dichlorobenzene | MS |
| n-dodecane | MS | 1,2-dichlorobenzene | MS |
| 1-hexene | MS | 1,2,4-trichlorobenzene | MS |
| 1,3-butadiene | MS | 1,1,2,3,4,4-hexachloro-1,3-butadiene | MS |
| 1,1-dichloroethene | MS | acrolein | MS |
| benzene | MS | acetone | MS |
| toluene | MS | 2-butanone | MS |
| m/p-xylene | MS | 2-propanol | MS |
| ethylbenzene | MS | 2-methoxy-2-methylpropane | MS |
| styrene | MS | vinylacetate | MS |
| o-xylene | MS | ethylacetate | MS |
| iso-propylbenzene | MS | tetrahydrofuran | MS |
| n-propylbenzene | MS | methyl methacrylate | MS |
| o-ethyltoluene | MS | 1,4-dioxane | MS |
| m-ethyltoluene | MS | 4-methyl-2-pentanone | MS |
| 1,2,3-trimethylbenzene | MS | 2-hexanone | MS |

The authors should include a more detailed description of their measurements different VOC classes and if some VOCs where only calculated as secondary products in the observation-based model (OBM) then this should be mentioned and the corresponding uncertainties should be discussed. If they were directly measure, it would be better to include OVOCS in the analysis of Figure 6.

**Response:** Thank you for your suggestions. We have added more details of the description of the VOC measurements in the manuscript, as the above question shown. The instrument system can quantitatively analyze 106 VOCs in the ambient atmosphere as shown in Table R1, but the observation-based model has high requirements on the quality of monitor data, so VOCs with concentrations below the detection line were removed. Therefore, the VOCs in Table S2 were the all VOCs compounds that we used in the OBM model in our study, and these valid data contained key species used in most relevant studies, which could constrain the

model and realize the localization of the model well (Liu et al., 2021; Chen et al., 2020; Xue et al., 2014).

About Figure 6: Firstly, it is RIRs of first-generation precursors for PAN. Secondly, the RIRs of measured OVOCs were very minor, which could be ignored in Figure 6. And the detailed information of some VOCs was added in the manuscript, as follows:

"The second-generation precursors of PAN of $CH_3CHO$ and MGLY have both primary and secondary sources, and the other OVOCs are mainly oxidation products of hydrocarbons (Sinha et al., 2019; Sarkar et al., 2017). Consequently, the contribution and importance of first-generation precursors of PAN are necessary to identify to better control photochemical pollution, which will be discussed in the next section".

"During these simulations (except for NO and $NO_2$), the model was not constrained by the OVOC measurements considering that these first-generation precursors contribute to PAN production through formation of OVOCs".

About the uncertainties of the model simulation results, the index of agreement (IOA) can be used to judge the reliability of the model simulation results, and the model validation results were added in the supplementary material of Text 1.

**Text 1 Model validation**

The index of agreement (IOA) can be used to judge the reliability of the model simulation results, and its equation is (Liu et al., 2019):

$$IOA = 1 - \frac{\sum_{i=1}^{n}(O_i - S_i)^2}{\sum_{i=1}^{n}(|O_i - \bar{O}| - |S_i - \bar{O}|)^2} \qquad (4)$$

where $S_i$ is simulated value, $O_i$ represents observed value, $\bar{O}$ is the average observed values, and n is the sample number. The IOA range is 0-1, and the higher the IOA value is, the better agreement between simulated and observed values is. In many studies, when IOA ranges from 0.68 to 0.90 (Wang et al., 2018), the simulation results are reasonable, and the IOA in our research is 0.88. Hence, the performance of the OBM-MCM model was reasonably acceptable.

**Table S2. Descriptive statistics of measured VOCs mixing ratios (Units: ppbv).**

| Chemicals | Spring | | Autumn | | Chemicals | Spring | | Autumn | |
|---|---|---|---|---|---|---|---|---|---|
| | Mean | SD | Mean | SD | | Mean | SD | Mean | SD |
| **Alkanes** | **9.41** | **5.30** | **5.47** | **2.88** | **Alkyne** | **1.00** | **0.55** | **0.63** | **0.34** |
| Ethane | 2.39 | 1.02 | 1.31 | 0.43 | **Aromatics** | **2.71** | **2.33** | **1.62** | **1.15** |
| Propane | 2.31 | 1.29 | 1.19 | 0.58 | Benzene | 0.27 | 0.14 | 0.16 | 0.09 |
| iso-Butane | 0.87 | 0.57 | 0.52 | 0.37 | Toluene | 1.37 | 1.21 | 0.85 | 0.84 |
| n-Butane | 1.30 | 0.94 | 0.77 | 0.59 | m/p-Xylene | 0.53 | 0.63 | 0.39 | 0.32 |
| iso-Pentane | 1.15 | 1.27 | 0.52 | 0.44 | Ethylbenzene | 0.18 | 0.18 | 0.09 | 0.10 |
| n-Pentane | 0.44 | 0.42 | 0.24 | 0.21 | Styrene | 0.09 | 0.16 | 0.02 | 0.04 |
| 2,2-Dimethylbutane | 0.02 | 0.02 | 0.02 | 0.01 | o-Xylene | 0.19 | 0.23 | 0.04 | 0.09 |
| 2,3-Dimethylbutane | 0.05 | 0.06 | 0.05 | 0.05 | m-Ethyltoluene | 0.02 | 0.02 | 0.01 | 0.01 |
| 2-Methylpentane | 0.08 | 0.09 | 0.05 | 0.04 | 1,3,5-Trimethylbenzene | 0.01 | 0.01 | 0.01 | 0.01 |
| 3-Methylpentane | 0.14 | 0.15 | 0.06 | 0.06 | p-Ethyltoluene | 0.01 | 0.01 | 0.01 | 0.005 |
| n-Hexane | 0.20 | 0.25 | 0.10 | 0.20 | 1,2,4-Trimethylbenzene | 0.03 | 0.05 | 0.01 | 0.02 |
| Cyclohexane | 0.04 | 0.04 | 0.02 | 0.02 | 1,2,3-Trimethylbenzene | 0.01 | 0.01 | 0.01 | 0.004 |
| 2-Methylhexane | 0.05 | 0.06 | 0.04 | 0.05 | **Isoprene (BHC)** | **0.08** | **0.14** | **0.10** | **0.17** |
| 3-Methylhexane | 0.08 | 0.09 | 0.05 | 0.08 | **Halocarbons** | **2.54** | **1.27** | **1.95** | **0.90** |
| n-Heptane | 0.07 | 0.08 | 0.05 | 0.06 | Chloromethane | 0.51 | 0.23 | 0.46 | 0.18 |
| n-Octane | 0.04 | 0.06 | 0.09 | 0.06 | Bromomethane | 0.06 | 0.03 | 0.04 | 0.02 |

| | | | | | | | | | |
|---|---|---|---|---|---|---|---|---|---|
| n-Nonane | 0.02 | 0.01 | 0.01 | 0.005 | Dichloromethane | 1.19 | 0.81 | 0.87 | 0.50 |
| n-Decane | 0.01 | 0.01 | 0.01 | 0.01 | Trichloromethane | 0.07 | 0.03 | 0.05 | 0.02 |
| n-Undecane | 0.02 | 0.02 | 0.03 | 0.03 | 1,2-Dichloroethane | 0.51 | 0.34 | 0.36 | 0.22 |
| n-Dodecane | 0.12 | 0.29 | 0.36 | 0.84 | Trichloroethene | 0.02 | 0.02 | 0.02 | 0.01 |
| **Alkenes** | **1.30** | **0.89** | **0.85** | **0.48** | 1,2-Dichloropropane | 0.12 | 0.13 | 0.10 | 0.08 |
| Ethene | 0.90 | 0.65 | 0.51 | 0.34 | Tetrachloroethene | 0.05 | 0.05 | 0.04 | 0.05 |
| Propene | 0.20 | 0.14 | 0.19 | 0.11 | **OVOCs** | **4.49** | **1.83** | **4.17** | **2.57** |
| 1-Butene | 0.04 | 0.03 | 0.03 | 0.02 | Acrolein | 0.06 | 0.03 | 0.04 | 0.02 |
| cis-2-Butene | 0.05 | 0.06 | 0.03 | 0.03 | Acetone | 2.22 | 0.94 | 2.21 | 0.91 |
| trans-2-Butene | 0.03 | 0.06 | 0.03 | 0.02 | 2-Butanone | 0.67 | 0.45 | 0.50 | 0.44 |
| 1-Pentene | 0.02 | 0.02 | 0.01 | 0.01 | 2-Propanol | 0.24 | 0.31 | 0.12 | 0.12 |
| trans-2-Pentene | 0.04 | 0.04 | 0.04 | 0.02 | 2-Methoxy-2-methylpropane | 0.24 | 0.32 | 0.09 | 0.09 |
| 1,3-Butadiene | 0.01 | 0.02 | 0.01 | 0.01 | Ethylacetate | 1.07 | 0.83 | 1.20 | 1.31 |

Minor comments:

Line 28: I could not quite understand what the authors intend to say in the following sentence "Without considering the transformation of peroxyacetyl radical (PA) and PAN, acetaldehyde contributed to the dominant production of PA (46±4%), followed by methylglyoxal (28±3%) and radical cycling (19±3%)."

**Response:** Thank you for your suggestions, and we are sorry for the confused expressions. The thermal decomposition of PAN can produce $NO_2$ and PA radicals (Eq. R1), and PA can quickly react with $NO_2$ producing PAN (Eq. R2). The ratio of reaction rates of $K_1$ and $K_2$ was close to 1, thus the contribution of PAN thermal decomposition to PA was minor. For better understanding, we revised the sentence as "Model simulations revealed that acetaldehyde oxidation (46±4%) contributed to the dominant formation pathway of PA (hence PAN), followed by methylglyoxal oxidation (28±3%) and radical cycling (19±3%)".

$$\text{The reaction rate of } K_1: \quad PAN \rightarrow PA + NO_2 \quad \text{(R1)}$$

$$\text{The reaction rate of } K_2: \quad PA + NO_2 \rightarrow PAN \quad \text{(R2)}$$

Line 30: Gramar needs to be improved, maybe the following will convey the intended meaning better: "The PAN formation was highly VOC-sensitive, as surplus NOx (compared with VOCs abundance) prevented NOx from being the limiting factor photochemical formation of secondary pollution."

**Response:** Thank you for your suggestions, we strongly agree with your descriptions, and we have revised this sentence as you said.

"The PAN formation was highly VOC-sensitive, as surplus NOx (compared with VOCs abundance) prevented NOx from being the limiting factor photochemical formation of secondary pollution".

Line 32: This sounds like a generic statement "PAN could promote or inhibit $O_3$ formation under high or low ROx levels, respectively.". It may be more appropriate to target this at the results of the present study "At our site, PAN promoted and inhibited $O_3$ formation under high and low ROx levels, respectively."

**Response:** Thank you for your suggestions, your expression was appropriate, and we revised the sentence

accordingly.

"At our site, PAN promoted and inhibited $O_3$ formation under high and low ROx levels, respectively".

Line 36: The authors could be a bit more assertive and specific in highlighting the contribution of their study to the scientific understanding. "Might be helpful" doesn't sound very convincing to me and doesn't specify the main contribution.

**Response:** Thank you for your suggestions. Your suggestions are pretty good, and we have replaced some words and emphasized the main contribution.

"The analysis of PAN formation mechanism and its positive or negative effect on ozone in our study provided scientific insights into photochemical pollution mechanism under various pollution scenarios in coastal areas".

Line 52: Gramar: "is the only formation pathway" instead of "is solely formation pathway"

**Response:** Thank you for your suggestions, we revised the sentence accordingly.

Line 72 Language: "were the most significant contributors" instead of "offered the highest contribution"

**Response:** Thank you for your suggestions, we have revised the expressions accordingly.

Line 74: "Recently, negative and positive impacts of PAN photochemistry on $O_3$ production were captured under the low and high NOx conditions, respectively." This statement should include a reference to the corresponding study.

**Response:** Thank you for your suggestions, we have added references to the corresponding study.

*Zeng, L., Fan, G. J., Lyu, X., Guo, H., Wang, J. L., and Yao, D.: Atmospheric fate of peroxyacetyl nitrate in suburban Hong Kong and its impact on local ozone pollution, Environ Pollut, 10.1016/j.envpol.2019.06.004, 2019.*

*Liu, Y., Shen, H., Mu, J., Li, H., Chen, T., Yang, J., Jiang, Y., Zhu, Y., Meng, H., Dong, C., Wang, W., and Xue, L.: Formation of peroxyacetyl nitrate (PAN) and its impact on ozone production in the coastal atmosphere of Qingdao, North China, Sci Total Environ, 778, 146265, 10.1016/j.scitotenv.2021.146265, 2021.*

Line 99: "was attributed to the downwind region of the downtown (Xiamen island) with densely population " can be simplified to "was downwind of the densely populated downtown region (Xiamen island)"

**Response:** Thank you for your suggestions, we have simplified the description.

Line 100 in my opinion field observation should be plural "field observations were" not singular

**Response:** Thank you for your suggestions, we agree with your opinion and changed its format into plural.

Line 122 I have never come across the term "ultrasonic atmospherium" before. I believe the correct name would be "weather station with 2D sonic anemometer".

**Response:** Thank you for your suggestions. We are sorry for the wrong usage of the term, and the correct name you mentioned was right. Hence, we changed the name to "weather station with sonic anemometer".

Line 209 please avoid colloquial language "The wind directions in late spring and early autumn were messy due to the season switch."  More scientific "During the transition from spring to summer the wind direction fluctuated between … and … while during the transition from summer to autumn the wind direction fluctuated from … to …"

**Response:** Thank you for your suggestions. We have revised this description as "During the transition from spring to summer the wind direction fluctuated between northwest and southeast while during the transition from summer to autumn the wind direction fluctuated from southeast to northeast".

Line 210, respectively missing at the end of the sentence "The wind rose charts showed that the wind direction frequencies with relatively high wind speed in spring and autumn were southeast wind and northeast wind (Fig. S2), respectively." Also define "high" by inserting a number "(>… m/s)" in brackets.

**Response:** Thank you for your suggestions. We have revised this sentence as "The wind rose charts showed that the wind direction frequencies with relatively high wind speed ($>3$ m·s$^{-1}$) in spring and autumn were southeast wind and northeast wind (Fig. S2), respectively".

In general, the manuscript should be run through a grammar check software before resubmission. It is better to avoid colloquial language and indirect phrases. It is OK to be direct and use simple sentences.

**Response:** We have performed the grammar corrections of our manuscript through grammar check software, and we also invited native speakers in related fields to polish the manuscript. Thanks for your suggestion, simple sentences, and direct phrases make my point clear.

**Reference:**

Huang, Y., Gao, S., Wu, S., Che, X., Yang, Y., Gu, J., Tan, W., Ruan, D., Xiu, G., and Fu, Q.: Stationary monitoring and source apportionment of VOCs in a chemical industrial park by combining rapid direct-inlet MSs with a GC-FID/MS, Sci Total Environ, 795, 148639, 10.1016/j.scitotenv.2021.148639, 2021.

Yang, Y., Ji, D., Sun, J., Wang, Yinghong, Yao, D., Zhao, S., Yu, X., Zeng, L., Zhang, R., Zhang, H., Wang, Y., Wang, Y.: Ambient volatile organic compounds in a suburban site between Beijing and Tianjin: concentration levels, source apportionment and health risk assessment. Sci. Total Environ. 695, 133889. https://doi.org/ 10.1016/j.scitotenv.2019.133889, 2019.

Qian, X., Shen, H., and Chen, Z.: Characterizing summer and winter carbonyl compounds in Beijing atmosphere, Atmospheric Environment, 214, 116845, 10.1016/j.atmosenv.2019.116845, 2019.

Liu, Y., Shen, H., Mu, J., Li, H., Chen, T., Yang, J., Jiang, Y., Zhu, Y., Meng, H., Dong, C., Wang, W., and Xue, L.: Formation of peroxyacetyl nitrate (PAN) and its impact on ozone production in the coastal atmosphere of Qingdao, North China, Sci Total Environ, 778, 146265, 10.1016/j.scitotenv.2021.146265, 2021.

Xue, L., Wang, T., Wang, X., Blake, D.R., Gao, J., Nie, W., et al.: On the use of an explicit chemical mechanism to dissect peroxy acetyl nitrate formation. Environ. Pollut. 195, 39–47, 2014.

Liu, L., Wang, X., Chen, J., Xue, L., Wang, W., Wen, L., Li, D., and Chen, T.: Understanding unusually high levels of peroxyacetyl nitrate (PAN) in winter in Urban Jinan, China, J Environ Sci (China), 71, 249-260, 10.1016/j.jes.2018.05.015, 2018.

Hu, B., Liu, T., Hong, Y., Xu, L., Li, M., Wu, X., Wang, H., Chen, J., and Chen, J.: Characteristics of peroxyacetyl nitrate (PAN) in a coastal city of southeastern China: Photochemical mechanism and pollution process, Sci Total Environ, 719, 137493, 10.1016/j.scitotenv.2020.137493, 2020.

Liu, T., Hong, Y., Li, M., Xu, L., Chen, J., Bian, Y., Yang, C., Dan, Y., Zhang, Y., Xue, L., Zhao, M., Huang, Z., and Wang, H.: Atmospheric oxidation capacity and ozone pollution mechanism in a coastal city of southeastern China: analysis of a typical photochemical episode by an observation-based model, Atmos. Chem. Phys., 22, 2173–2190, https://doi.org/10.5194/acp-22-2173-2022, 2022.

Ma Y, Ma B, Jiao H, Zhang Y, Xin J, Yu Z: An analysis of the effects of weather and air pollution on tropospheric ozone using a generalized additive model in Western China: Lanzhou, Gansu. Atmos. Environ., 224:117342, 2020.

Yan, R. E., Ye, H., Lin, X., He, X., Chen, C., Shen, J.D., Xu, K.E., Zhen, X. Y., Wang, L. J.: Characteristics and influence factors of ozone pollution in Hangzhou. Acta Sci. Circumstantiae, 38 (3), 1128–1136, 2018.

Zeng, L. W., Fan, G. J., Lyu, X. P., Guo, H., Wang, J. L., Yao, D. W.: Atmospheric fate of peroxyacetyl nitrate in suburban Hong Kong and its impact on local ozone pollution. Environ. Pollut. 252, 1910–1919, 2019.

Wu, X., Li, M., Chen, J., Wang, H., Xu, L., Hong, Y., Zhao, G., Hu, B., Zhang, Y., Dan, Y., and Yu, S.: The characteristics of air pollution induced by the quasi-stationary front: Formation processes and influencing factors, Sci Total Environ, 707, 136194, 10.1016/j.scitotenv.2019.136194, 2020.

Liu, T., Hu, B., Xu, X., Hong, Y., Zhang, Y., Wu, X., Xu, L., Li, M., Chen, Y., Chen, X., and Chen, J.: Characteristics of PM2.5-bound secondary organic aerosol tracers in a coastal city in Southeastern China: Seasonal patterns and pollution identification, Atmospheric Environment, 237, 117710, 10.1016/j.atmosenv. 2020.117710, 2020a.

Liu, T., Hu, B., Yang, Y., Li, M., Hong, Y., Xu, X., Xu, L., Chen, N., Chen, Y., Xiao, H., and Chen, J.: Characteristics and source apportionment of PM2.5 on an island in Southeast China: Impact of sea-salt and monsoon, Atmospheric Research, 235, 104786, 10.1016/j.atmosres.2019.104786, 2020b.

Chen, T., Xue, L., Zheng, P., Zhang, Y., Liu, Y., Sun, J., Han, G., Li, H., Zhang, X., Li, Y., Li, H., Dong, C., Xu, F., Zhang, Q., and Wang, W.: Volatile organic compounds and ozone air pollution in an oil production region in northern China, Atmospheric Chemistry and Physics, 20, 7069-7086, 10.5194/acp-20-7069-2020, 2020.

Wang, Y., Guo, H., Zou, S., Lyu, X., Ling, Z., Cheng, H., and Zeren, Y.: Surface O3 photochemistry over the South China Sea: Application of a near-explicit chemical mechanism box model, Environ. Pollut., 234, 155–166, https://doi.org/10.1016/j.envpol.2017.11.001, 2018.

Liu, J., Wang, L., Li, M., Liao, Z., Sun, Y., Song, T., Gao, W., Wang, Y., Li, Y., Ji, D., Hu, B., Kerminen, V.-M., Wang, Y., and Kulmala, M.: Quantifying the impact of synoptic circulation patterns on ozone variability in northern China from April to October 2013–2017, Atmos. Chem. Phys., 19, 14477–14492, https://doi.org/10.5194/acp-19-14477-2019, 2019.

Pallavi, Sinha, B., and Sinha, V.: Source apportionment of volatile organic compounds in the northwest Indo-Gangetic Plain using a positive matrix factorization model, Atmos. Chem. Phys., 19, 15467–15482, https://doi.org/10.5194/acp-19-15467-2019, 2019.

Sarkar, C., Sinha, V., Sinha, B., Panday, A. K., Rupakheti, M., and Lawrence, M. G.: Source apportionment of NMVOCs in the Kathmandu Valley during the SusKat-ABC international field campaign using positive matrix factorization, Atmos. Chem. Phys., 17, 8129-8156, https://doi.org/10.5194/acp-17-8129-2017, 2017.

---

## Author Response (AR3)

**Response to Reviewers**

Comment on acp-2021-948

**RC1 Anonymous Referee #1**

Comments on "Seasonal characteristics of atmospheric peroxyacetyl nitrate (PAN) in a coastal city 2 of Southeast China: Explanatory factors and photochemical effects" by Liu et al

The manuscript reports influencing factors to PAN pollution in China. The manuscript reports important results. It is suitable for the Journal Atmospheric Chemistry and Physics. I suggest authors incorporate the below suggestions before its publication.

**Response:** Thank you very much for your exploratory and constructive advice. Here, we have carefully revised the manuscript.

Major comments:

• In the abstract section, the authors state that the current paper reports the formation mechanism of PAN and its effect on ozone were identified. I suggest the authors explain it in brief.

**Response:** Thank you for your suggestions. PAN can be produced only through the reaction of $PA+NO_2$ (Liu et al., 2021; Xue et al., 2014). Hence, the production and sink of PA can represent that of the PAN mechanism indirectly. In the abstract section, we have explained the PA formation mechanism (hence PAN formation mechanism). The effect of PAN on ozone represented that PAN could promote or inhibit $O_3$ formation under high or low ROx levels, respectively. For better understanding, we revised the relevant expression, as follows:

"Model simulations revealed that acetaldehyde oxidation ($46\pm4\%$) contributed to the dominant formation pathway of PA (hence PAN), followed by methylglyoxal oxidation ($28\pm3\%$) and radical cycling ($19\pm3\%$)".

"The analysis of PAN formation mechanism and its positive or negative effect on ozone provided scientific insights into photochemical pollution mechanism under various pollution scenarios in coastal areas".

• In section 2.2, some details of the box model should be added here, although it is explained in the previous study, e.g., details of computation of net production rate of $O_3$.

**Response:** Thank you for your suggestions. More details about the box model have been added, as follows:

"The observed data with a time resolution of 1 h of pollutants (i.e., $O_3$, CO, NO, $NO_2$, HONO, $SO_2$, and VOCs), meteorological parameters (i.e., T, P, and RH), and photolysis rate constants ($J(O^1D)$, $J(NO_2)$, $J(H_2O_2)$, $J(HONO)$, $J(HCHO)$, and $J(NO_3)$), which were mentioned in Section 2.1, were input into the OBM-MCM model as constraints".

"The production pathways of $O_3$ include $HO_2+NO$ and $RO_2+NO$ reactions, and the destruction pathways of $O_3$ involve reactions of $O_3$ photolysis, $O_3+OH$, $O_3+HO_2$, $O_3+VOCs$, $NO_2+OH$, and $NO_3+VOCs$. The net $O_3$ production rate ($P(O_3)$) is calculated by the difference of $O_3$ production rate and destruction rate".

- The study used a non-parametric regression model. How good is it? Have you compared it with traditional chemistry models? Have you compared results with them?

**Response:** Thank you for your suggestions. We have added the validation of the GAM model in the supplementary material of Text 1, as follows.

**Text 1 Model Validation.**

Figure T1 and T2 show the residual test results of the Generalized Additive Model (GAM) in spring and autumn, respectively. From the residual Q-Q plots (Fig. T1 (a) and Fig. T2 (a)), the points were mostly on a straight line, indicating that the residuals conformed to a normal distribution. Meanwhile, the residual histogram of the model in Fig. T1 (c) and Fig. T2 (c) showed that the residuals were mainly concentrated around 0, which demonstrated the good fitting degree of the model. From the scatter plot of residuals and linear prediction values (Fig. T1 (b) and Fig. T2 (b)), the residuals were randomly distributed. From the scatter plot of the observed values and the fitted values (Fig. T1 (d) and Fig. T2 (d)), the response variables and the fitted values were well matched, and basically showed a "y = x" distribution. Therefore, the fitting effect of this model was good.

The function of the Generalized Additive Models (GAM) is to analyze the correlation between the explanatory variables and the response variables. The GAM has been widely used in air pollution research and can effectively deal with the complex nonlinear relationship between air pollutants and influencing factors. Although GAM offers a flexible approach to calculating trends, the model is just a regression statistical model/method, which could not establish a connection with traditional chemistry models. Hence, comparing results with them might be very difficult and less significance.

[Figure]

**Fig. T1 Residual test results of the Generalized Additive Model (GAM) in spring.**

[Figure]

**Fig. T2 Residual test results of the Generalized Additive Model (GAM) in autumn.**

- Section 3.2, L226-229, statements are not clear. The statement "PAN pollution was mainly from local production" should be explained.

**Response:** Thank you for your suggestions, and we're sorry for the unclear expressions. We have revised the relevant contents in our manuscript.

"High PAN values in spring easily happened in the wind direction of the southeast with low wind speed (<3 m·s$^{-1}$), showing the influence of urban plumes from the downtown of Xiamen island. High PAN values in autumn also appeared in the wind direction of the southeast, as well as the northeast with a relatively high wind speed (from Quanzhou city, an industrial city adjacent to Xiamen). Anymore, PAN lifetimes in our observation site were relatively short due to the high ambient temperature, and the PAN lifetimes in autumn (2.02 hours) were significantly lower than that in spring (6.39 hours), which was not conducive to regional transport (Hu et al., 2020; Liu et al., 2018). Accordingly, O$_3$ showed obvious characteristics of long-range transport, and PAN pollution was mainly from local production/accumulation in spring and autumn, but short-range transport from adjacent cities might contribute to the high PAN concentrations in autumn to a certain extent".

- L230-235, it is not clear how local mixing is computed.

**Response:** Thank you for your suggestions. The mixing ratio in our study represents the volume concentration, which could be acquired directly from monitoring instruments. The mixing ratio of a gas component refers to

the occupied volume ratio of this gas component in dry air under the same temperature and pressure conditions. Mixing ratio values strictly speaking may not need a unit, but commonly ppb, nmol/mol or similar would be used. In our study, the units of the monitored pollutants (such as $O_3$, PAN, and VOCs) were ppbv. Hence, we named their volume concentration as mixing ratio. For better understanding, we have changed the expression in this part. The detailed modifications are as follows:

"Based on the above analysis, we found that the photochemical reactions were still intense and even stronger under the low precursor levels. Although the precursor abundances of PAN and $O_3$ in spring were significantly higher than those in autumn (P<0.01), PAN values were comparable to and $O_3$ values were much higher in autumn than those in spring, respectively".

- Fig. 3 and Fig. S4, nonlinear relations between variables (NO, UV, RH, T Ox) and PAN are well known. What is new in these figures should be explained.

**Response:** Thank you for your suggestions, we rewrote this part and added some new discussions in these figures. The details are as follows:

"The PAN in both seasons showed a downward trend with the increase of NO. PAN in spring was constant with NO fluctuation between 10 and 23 ppbv, and the confidence interval (CI) of NO concentration was relatively narrow. As we all know, the reaction of PA+NO is one of the most important loss pathways of PA, and the $NO_2$ production by NO oxidation in the $O_3$ formation cycle can react with PA radical to produce PAN, suggesting the fact that NO can consume and produce PAN indirectly (Liu et al., 2021). The consumption of NO to PAN was basically equal to the production when the NO levels were relatively high (>10 ppbv), and the consumption of NO to PAN is greater than the production when the NO levels were low in spring. High values of NO mainly happened during rush hour traffic, thus controlling vehicle emissions can effectively alleviate PAN pollution. Ox had a positive correlation with PAN, representing the promotion effects of atmospheric oxidation capacity on PAN formation. The Ox levels <70 ppbv (with narrow CI) played a significant promotion role in PAN formation (Fig. 3(b) and Fig. S4(b)). High Ox >70 ppbv showed little influence on PAN, which could be explained as high Ox with relatively high air temperature leading to intense PAN thermal decomposition. When TVOCs were between 10 and 30 ppbv and $PM_{2.5}$ levels were <17 $\mu g \cdot m^{-3}$, PAN showed an upward trend with narrow CI. According to our previous study (Liu et al., 2022; Hu et al., 2020), the results of sensitivity analysis in Xiamen was VOCs-sensitive; the relatively low $PM_{2.5}$ concentrations in Xiamen showed limited influence on solar radiation through scattering and absorption, but promoted heterogeneous reactions producing radicals to a certain extent. UV and T had significant positive and negative nonlinear correlations with PAN, respectively. When UV changed between 0 and 50 $W \cdot m^{-2}$ and T changed between 15 and 35 $W \cdot m^{-2}$, the CIs barely increased. In addition, when RH was more than 40%, the increase of RH was unfavorable for PAN production in both seasons. Some studies also found that high water vapor content could remove PAN and its precursors (Yan et al., 2018; Ma et al., 2020). Overall, the multiple-factor GAM analysis could better simulate the variations of PAN under real atmospheric conditions and evaluate the contributions of the influence factors to PAN formation".

- L261-263: The solar radiations are stronger in spring than autumn; hence UV, T, and OX will be more effective in PAN formation during spring and vice-a-versa in autumn. Why should you mention it explicitly?

**Response:** Thank you for your suggestions, we agree that this summary was not accurate enough. In the previous question of Fig. 3 and Fig. S4, we rewrote this section and added some new information. Hence, we have rewritten this summary for a new version to avoid unnecessary misunderstandings.

"Overall, the multiple-factor GAM analysis could better simulate the variations of PAN under real atmospheric conditions and evaluate the contributions of the influence factors to PAN formation".

- L292-302. These statements are huge. It is unclear whether they are supported by the model simulations or observations.

**Response:** We agree with your suggestions, and we rewrote these statements based on observations. The revisions in the manuscript are as follows.

"Diurnal variations of air pollutants during episodes and non-episodes are shown in Fig. 4, which could be explained by the evolution of the planetary boundary layer, local emissions, and atmospheric photochemistry. PAN reached a maximum value at 12:00-14:00, then decreased with weak solar radiation and reached the lowest in the early morning. Similar diurnal patterns of PAN and $O_3$ were observed, indicating the dominance of local photochemistry during the observation period (Zeng et al., 2019). CO, NOx and TVOCs showed highest values in the morning and the lowest values in the afternoon".

- L325-328: All possible factors, meteorological conditions, accumulation of pollution, local transport, etc., are mentioned here. What is the most influencing factor?

**Response:** Thank you for your suggestions, we are sorry for the complicated and indirect expression of this sentence. We have revised the sentence as "These results indicated favorable meteorological condition was the dominant factor to produce PAN through accelerating its production rate and accumulation".

Section 3.4 is lengthy but informative. It should be divided into two sections.

**Response:** Thank you for your suggestions, we have divided this section into two sections (Section 3.4.1 Inhibition and promotion effect of PAN on $O_3$ formation; Section 3.4.2 The influencing factors during inhibition and promotion stages).

Minor comments:

- The quality of figure 7 should be improved.

**Response:** Thank you for your suggestions, and we have improved the quality of figure 7.

[Figure]

**Fig. 7. The differences of O₃ net production △P(O₃), △OH, △HO₂, △RO₂, △NO and △NO₂ between the SC1 and the SC2 during the daytime (06:00-17:00) in (a) spring and (b) autumn (Unit: ppbv·h⁻¹ for △P(O₃); ppbv for △NO and △NO₂; molecules·cm⁻³ for △OH, △HO₂ and △RO₂). The SC1 scenario was the base scenario putting all detected data (i.e. VOCs, trace gases, and meteorological parameters) into the model with all reaction pathways of the MCM mechanism, and the SC2 disabled the PAN chemistry, which is the only difference between SC1 and SC2.**

- Section 2.1 can be moved to the supplementary material. I have difficulty in reading the x-axis labels in Figure 1.

**Response:** Thanks for your suggestion. We have changed a high resolution picture. Considering the importance of observation experiment details, we have simplified this part and divided them into two sections (Section 2.1 Observation site; Section 2.2 Measurement techniques). About Figure 1, the specific changes are as follows.

**2.1 Observation site**

[Figure]

**Fig. 1. Location of Xiamen and the observation site.**

[revised manuscript text omitted]

**RC2 Anonymous Referee #2**

The paper entitled "Seasonal characteristics of atmospheric peroxyacetyl nitrate (PAN) 1 in a coastal city 2 of Southeast China: Explanatory factors and photochemical effects" is generally well written, interesting and within the scope of ACP. However, the authors need to address a few minor comments, before the paper is suitable to be published in ACP.

**Response:** Thanks for your valuable comments. We have corrected this manuscript according to your suggestions.

Major comments:

I find the description of the VOC measurements a bit brief. Which classes of VOCs did the authors monitor with their GC and using which columns? It is not clear whether aldehydes and ketones were measured and if so which ones. Figure S5 doesn't discuss OVOCs as ozone precursors. While table S2 includes some OVOCs, it doesn't mention some important compounds like acetaldehyde, MVK, methylglyoxal and methacrolein. Where these compounds measured and if yes how? It appears to me that the authors are assuming that these are primarily secondary compounds that are derived from other primary emissions ethane, propene, isoprene and aromatics (line 57). If they are just model calculated and not measured then this could bias the results. While these compounds, no doubt have photochemical sources, there can also be substantial primary

emissions particularly from various types of biomass burning (Pallavi et al. https://doi.org/10.5194/acp-19-15467-2019), industrial sources and, in the case of acetaldehyde also vegetation (Sarkar et al. 2017 https://doi.org/10.5194/acp-17-8129-2017)

**Response:** Thank you for your suggestions. We have added more details of the description of the VOC measurements in the manuscript, as follows. Meanwhile, the detailed compounds of 106 VOCs were shown in Table R1.

"A gas chromatography-mass spectrometer (GC-FID/MS, TH-300B, Wuhan, CN) was used for monitoring the atmospheric VOCs with a 1-hour time resolution. The instrument conducted sampling with a 30 L/min sampling rate, then samples were pre-concentrated by cooling to -160 °C in a cryogenic trap followed by heating to 100 °C, and subsequently transferred to the secondary trap by high-purity helium (He). The flame ionization detector (FID) detected the low-carbon (C2-C5) hydrocarbons by a PLOT ($Al_2O_3$/KCl) column (15 m × 0.32 mm × 6.0 μm); the other compounds were quantified using a DB-624 column (60 m × 0.25 mm × 1.4 μm). The instrument system can quantitatively analyze 106 VOCs in the ambient atmosphere, including 29 alkanes, 11 alkenes, one alkyne, 17 aromatics, 35 halogenated hydrocarbons, and 13 OVOCs. The single-point calibration was performed every day at 23:00 with the standard mixtures of PAMS and TO15, and multi-point calibration was performed one month. The detection limits of the measured VOCs were in the range of 0.02 ppbv to 0.30 ppbv, and the measurement precision was ≤10%".

Because the states of calibration gases of aldehydes and ketones are very unstable, thereby it is difficult to offer calibration gases during observation campaigns in our observation sites. Hence, the nine compounds can not be quantified (Acetaldehyde, Propanal, Crotonaldehyde, Methacrolein, n-butanal, Benzaldehyde, Valeraldehyde, m-Tolualdehyde, Hexanal). However, this phenomenon is common in many studies (Huang et al., 2021; Yang et al., 2019; Liu et al., 2021). Hence, the OVOCs that you mentioned can not be measured. All compounds of 106 VOCs detected by the gas chromatography-mass spectrometer are shown in Table R1. Meanwhile, the observation-based model has high requirements on the quality of the monitor data, and quantities of the pollutants inputting into the OBM model are limited, so VOCs with concentrations below the detection line were removed. Therefore, the VOCs in Table S2 were the all VOCs compounds that we used in the OBM model in our study.

The RIR values in Figure S5 were calculated by the observation data. As said above, some important aldehydes and ketones can not be considered, so the RIRs of OVOCs were underestimated. Hence, the discussion of RIRs of OVOCs is not included. In many relevant studies, their results of RIR also did not consider the OVOCs (Liu et al., 2021; Chen et al., 2020). In the future, we will calculate the RIR of OVOCs, because of our upgraded monitoring station with the OVOCs measurements.

Thank you for your suggestions. We agree with your opinions that the precursors of PAN of OVOCs have both primary and secondary sources (Sinha et al., 2019; Sarkar et al., 2017). When these OVOCs concentrations were not observed, the concentrations could be locally and reasonably calculated by the model. The three modules (initial value settings, observation data input, pre-ran for 2 days before running the OBM model) could constrain the unmeasured compounds reaching a steady-state and make the model localization. Some studies exploring the PAN formation mechanism based on OBM model also did not observe these OVOCs data (Chen et al., 2020, Liu et al., 2021; Xue et al., 2014). Meanwhile, we strongly agree with your idea and realized the importance of OVOCs, and some OVOCs of model calculation without measuring could bias the results. Hence, our team improved the monitoring of OVOCs in October 2021. A more optimized and complete monitoring system is also the future optimization goal of our model. Anymore, the index of agreement (IOA) judging the reliability of the model simulation results showed the performance of the OBM-MCM model was reasonably acceptable, and its detailed calculation is shown in the next question.

**Table R1 The detailed compounds of 106 VOCs and the detectors.**

| Compounds | Detector | Compounds | Detector |
|---|---|---|---|
| ethene | FID | p-ethyltoluene | MS |
| ethane | FID | 1,2,4-trimethylbenzene | MS |
| propane | FID | 1,2,3-trimethylbenzene | MS |
| iso-butane | FID | m-diethylbenzene | MS |
| n-butane | FID | p-diethylbenzene | MS |
| iso-pentane | FID | naphthalene | MS |
| n-pentane | FID | dichlorodifluoromethane | MS |
| cyclopentane | FID | chloromethane | MS |
| propene | FID | 1,1,2,2-tetrachloro-Ethane | MS |
| 1-butene | FID | vinylchloride | MS |
| cis-2-butene | FID | bromomethane | MS |
| trans-2-butene | FID | chloroethane | MS |
| 1-pentene | FID | trichlorofluoromethane | MS |
| trans-2-pentene | FID | 1,1,2-trichloro-1,2,2-trifluoroethane | MS |
| cis-2-pentene | FID | carbondisulfide | MS |
| acetylene | FID | dichloromethane | MS |
| isoprene | FID | cis-1,2-dichloroethene | MS |
| 2,2-dimethylbutane | MS | 1,1-dichloroethane | MS |
| 2,3-dimethylbutane | MS | trans-1,2-dichloroethene | MS |
| 2-methylpentane | MS | trichloromethane | MS |
| 3-methylpentane | MS | 1,1,1-trichloroethane | MS |
| n-hexane | MS | 1,2-dichloroethane | MS |
| 2,4-dimethylpentane | MS | carbon tetrachloride | MS |
| methylcyclopentane | MS | trichloroethene | MS |
| cyclohexane | MS | 1,2-dichloropropane | MS |
| 2-methylhexane | MS | dichlorobromomethane | MS |
| 2,3-dimethylpentane | MS | cis-1,3-dichloropropene | MS |
| 3-methylhexane | MS | trans-1,3-dichloropropene | MS |
| 2,2,4-trimethylpentane | MS | 1,1,2-trichloroethane | MS |
| n-heptane | MS | dibromochloromethane | MS |
| methylcyclohexane | MS | tetrachloroethene | MS |
| 2,3,4-trimethylpentane | MS | 1,2-ethylenedibromide | MS |
| 2-methylheptane | MS | chlorobenzene | MS |
| 3-methylheptane | MS | tribromomethane | MS |
| n-octane | MS | Ethane, 1,1,2,2-tetrachloro- | MS |
| n-nonane | MS | 1,3-dichlorobenzene | MS |
| n-decane | MS | chlorotoluene | MS |
| n-undecane | MS | 1,4-dichlorobenzene | MS |
| n-dodecane | MS | 1,2-dichlorobenzene | MS |
| 1-hexene | MS | 1,2,4-trichlorobenzene | MS |
| 1,3-butadiene | MS | 1,1,2,3,4,4-hexachloro-1,3-butadiene | MS |
| 1,1-dichloroethene | MS | acrolein | MS |
| benzene | MS | acetone | MS |
| toluene | MS | 2-butanone | MS |
| m/p-xylene | MS | 2-propanol | MS |
| ethylbenzene | MS | 2-methoxy-2-methylpropane | MS |
| styrene | MS | vinylacetate | MS |

| | | | |
|---|---|---|---|
| o-xylene | MS | ethylacetate | MS |
| iso-propylbenzene | MS | tetrahydrofuran | MS |
| n-propylbenzene | MS | methyl methacrylate | MS |
| o-ethyltoluene | MS | 1,4-dioxane | MS |
| m-ethyltoluene | MS | 4-methyl-2-pentanone | MS |
| 1,2,3-trimethylbenzene | MS | 2-hexanone | MS |

The authors should include a more detailed description of their measurements different VOC classes and if some VOCs where only calculated as secondary products in the observation-based model (OBM) then this should be mentioned and the corresponding uncertainties should be discussed. If they were directly measure, it would be better to include OVOCS in the analysis of Figure 6.

**Response:** Thank you for your suggestions. We have added more details of the description of the VOC measurements in the manuscript, as the above question shown. The instrument system can quantitatively analyze 106 VOCs in the ambient atmosphere as shown in Table R1, but the observation-based model has high requirements on the quality of monitor data, so VOCs with concentrations below the detection line were removed. Therefore, the VOCs in Table S2 were the all VOCs compounds that we used in the OBM model in our study, and these valid data contained key species used in most relevant studies, which could constrain the model and realize the localization of the model well (Liu et al., 2021; Chen et al., 2020; Xue et al., 2014).

About Figure 6: Firstly, it is RIRs of first-generation precursors for PAN. Secondly, the RIRs of measured OVOCs were very minor, which could be ignored in Figure 6. And the detailed information of some VOCs was added in the manuscript, as follows:

"The second-generation precursors of PAN of $CH_3CHO$ and MGLY have both primary and secondary sources, and the other OVOCs are mainly oxidation products of hydrocarbons (Sinha et al., 2019; Sarkar et al., 2017). Consequently, the contribution and importance of first-generation precursors of PAN are necessary to identify to better control photochemical pollution, which will be discussed in the next section".

"During these simulations (except for NO and $NO_2$), the model was not constrained by the OVOC measurements considering that these first-generation precursors contribute to PAN production through formation of OVOCs".

About the uncertainties of the model simulation results, the index of agreement (IOA) can be used to judge the reliability of the model simulation results, and the model validation results were added in the supplementary material of Text 1.

**Text 1 Model validation**

The index of agreement (IOA) can be used to judge the reliability of the model simulation results, and its equation is (Liu et al., 2019):

$$IOA = 1 - \frac{\sum_{i=1}^{n}(O_i-S_i)^2}{\sum_{i=1}^{n}(|O_i-\bar{O}|-|S_i-\bar{O}|)^2} \quad (4)$$

where $S_i$ is simulated value, $O_i$ represents observed value, $\bar{O}$ is the average observed values, and n is the sample number. The IOA range is 0-1, and the higher the IOA value is, the better agreement between simulated and observed values is. In many studies, when IOA ranges from 0.68 to 0.90 (Wang et al., 2018), the simulation results are reasonable, and the IOA in our research is 0.88. Hence, the performance of the OBM-MCM model was reasonably acceptable.

**Table S2. Descriptive statistics of measured VOCs mixing ratios (Units: ppbv).**

| Chemicals | Spring Mean | Spring SD | Autumn Mean | Autumn SD | Chemicals | Spring Mean | Spring SD | Autumn Mean | Autumn SD |
|---|---|---|---|---|---|---|---|---|---|
| **Alkanes** | **9.41** | **5.30** | **5.47** | **2.88** | **Alkyne** | **1.00** | **0.55** | **0.63** | **0.34** |
| Ethane | 2.39 | 1.02 | 1.31 | 0.43 | **Aromatics** | **2.71** | **2.33** | **1.62** | **1.15** |
| Propane | 2.31 | 1.29 | 1.19 | 0.58 | Benzene | 0.27 | 0.14 | 0.16 | 0.09 |
| iso-Butane | 0.87 | 0.57 | 0.52 | 0.37 | Toluene | 1.37 | 1.21 | 0.85 | 0.84 |
| n-Butane | 1.30 | 0.94 | 0.77 | 0.59 | m/p-Xylene | 0.53 | 0.63 | 0.39 | 0.32 |
| iso-Pentane | 1.15 | 1.27 | 0.52 | 0.44 | Ethylbenzene | 0.18 | 0.18 | 0.09 | 0.10 |
| n-Pentane | 0.44 | 0.42 | 0.24 | 0.21 | Styrene | 0.09 | 0.16 | 0.02 | 0.04 |
| 2,2-Dimethylbutane | 0.02 | 0.02 | 0.02 | 0.01 | o-Xylene | 0.19 | 0.23 | 0.04 | 0.09 |
| 2,3-Dimethylbutane | 0.05 | 0.06 | 0.05 | 0.05 | m-Ethyltoluene | 0.02 | 0.02 | 0.01 | 0.01 |
| 2-Methylpentane | 0.08 | 0.09 | 0.05 | 0.04 | 1,3,5-Trimethylbenzene | 0.01 | 0.01 | 0.01 | 0.01 |
| 3-Methylpentane | 0.14 | 0.15 | 0.06 | 0.06 | p-Ethyltoluene | 0.01 | 0.01 | 0.01 | 0.005 |
| n-Hexane | 0.20 | 0.25 | 0.10 | 0.20 | 1,2,4-Trimethylbenzene | 0.03 | 0.05 | 0.01 | 0.02 |
| Cyclohexane | 0.04 | 0.04 | 0.02 | 0.02 | 1,2,3-Trimethylbenzene | 0.01 | 0.01 | 0.01 | 0.004 |
| 2-Methylhexane | 0.05 | 0.06 | 0.04 | 0.05 | **Isoprene (BHC)** | **0.08** | **0.14** | **0.10** | **0.17** |
| 3-Methylhexane | 0.08 | 0.09 | 0.05 | 0.08 | **Halocarbons** | **2.54** | **1.27** | **1.95** | **0.90** |
| n-Heptane | 0.07 | 0.08 | 0.05 | 0.06 | Chloromethane | 0.51 | 0.23 | 0.46 | 0.18 |
| n-Octane | 0.04 | 0.06 | 0.09 | 0.06 | Bromomethane | 0.06 | 0.03 | 0.04 | 0.02 |
| n-Nonane | 0.02 | 0.01 | 0.01 | 0.005 | Dichloromethane | 1.19 | 0.81 | 0.87 | 0.50 |
| n-Decane | 0.01 | 0.01 | 0.01 | 0.01 | Trichloromethane | 0.07 | 0.03 | 0.05 | 0.02 |
| n-Undecane | 0.02 | 0.02 | 0.03 | 0.03 | 1,2-Dichloroethane | 0.51 | 0.34 | 0.36 | 0.22 |
| n-Dodecane | 0.12 | 0.29 | 0.36 | 0.84 | Trichloroethene | 0.02 | 0.02 | 0.02 | 0.01 |
| **Alkenes** | **1.30** | **0.89** | **0.85** | **0.48** | 1,2-Dichloropropane | 0.12 | 0.13 | 0.10 | 0.08 |
| Ethene | 0.90 | 0.65 | 0.51 | 0.34 | Tetrachloroethene | 0.05 | 0.05 | 0.04 | 0.05 |
| Propene | 0.20 | 0.14 | 0.19 | 0.11 | **OVOCs** | **4.49** | **1.83** | **4.17** | **2.57** |
| 1-Butene | 0.04 | 0.03 | 0.03 | 0.02 | Acrolein | 0.06 | 0.03 | 0.04 | 0.02 |
| cis-2-Butene | 0.05 | 0.06 | 0.03 | 0.03 | Acetone | 2.22 | 0.94 | 2.21 | 0.91 |
| trans-2-Butene | 0.03 | 0.06 | 0.03 | 0.02 | 2-Butanone | 0.67 | 0.45 | 0.50 | 0.44 |
| 1-Pentene | 0.02 | 0.02 | 0.01 | 0.01 | 2-Propanol | 0.24 | 0.31 | 0.12 | 0.12 |
| trans-2-Pentene | 0.04 | 0.04 | 0.04 | 0.02 | 2-Methoxy-2-methylpropane | 0.24 | 0.32 | 0.09 | 0.09 |
| 1,3-Butadiene | 0.01 | 0.02 | 0.01 | 0.01 | Ethylacetate | 1.07 | 0.83 | 1.20 | 1.31 |

Minor comments:

Line 28: I could not quite understand what the authors intend to say in the following sentence "Without considering the transformation of peroxyacetyl radical (PA) and PAN, acetaldehyde contributed to the dominant production of PA (46±4%), followed by methylglyoxal (28±3%) and radical cycling (19±3%)."

**Response:** Thank you for your suggestions, and we are sorry for the confused expressions. The thermal decomposition of PAN can produce $NO_2$ and PA radicals (Eq. R1), and PA can quickly react with $NO_2$ producing PAN (Eq. R2). The ratio of reaction rates of $K_1$ and $K_2$ was close to 1, thus the contribution of PAN thermal decomposition to PA was minor. For better understanding, we revised the sentence as "Model simulations revealed that acetaldehyde oxidation (46±4%) contributed to the dominant formation pathway of PA (hence PAN), followed by methylglyoxal oxidation (28±3%) and radical cycling (19±3%)".

The reaction rate of $K_1$: $\quad PAN \rightarrow PA + NO_2 \qquad$ (R1)

The reaction rate of $K_2$: $PA + NO_2 \rightarrow PAN$ (R2)

Line 30: Gramar needs to be improved, maybe the following will convey the intended meaning better: "The PAN formation was highly VOC-sensitive, as surplus NOx (compared with VOCs abundance) prevented NOx from being the limiting factor photochemical formation of secondary pollution."

**Response:** Thank you for your suggestions, we strongly agree with your descriptions, and we have revised this sentence as you said.

"The PAN formation was highly VOC-sensitive, as surplus NOx (compared with VOCs abundance) prevented NOx from being the limiting factor photochemical formation of secondary pollution".

Line 32: This sounds like a generic statement "PAN could promote or inhibit $O_3$ formation under high or low ROx levels, respectively.". It may be more appropriate to target this at the results of the present study "At our site, PAN promoted and inhibited $O_3$ formation under high and low ROx levels, respectively."

**Response:** Thank you for your suggestions, your expression was appropriate, and we revised the sentence accordingly.

"At our site, PAN promoted and inhibited $O_3$ formation under high and low ROx levels, respectively".

Line 36: The authors could be a bit more assertive and specific in highlighting the contribution of their study to the scientific understanding. "Might be helpful" doesn't sound very convincing to me and doesn't specify the main contribution.

**Response:** Thank you for your suggestions. Your suggestions are pretty good, and we have replaced some words and emphasized the main contribution.

"The analysis of PAN formation mechanism and its positive or negative effect on ozone in our study provided scientific insights into photochemical pollution mechanism under various pollution scenarios in coastal areas".

Line 52: Gramar: "is the only formation pathway" instead of "is solely formation pathway"

**Response:** Thank you for your suggestions, we revised the sentence accordingly.

Line 72 Language: "were the most significant contributors" instead of "offered the highest contribution"

**Response:** Thank you for your suggestions, we have revised the expressions accordingly.

Line 74: "Recently, negative and positive impacts of PAN photochemistry on $O_3$ production were captured under the low and high NOx conditions, respectively." This statement should include a reference to the corresponding study.

**Response:** Thank you for your suggestions, we have added references to the corresponding study.

*Zeng, L., Fan, G. J., Lyu, X., Guo, H., Wang, J. L., and Yao, D.: Atmospheric fate of peroxyacetyl nitrate in suburban Hong Kong and its impact on local ozone pollution, Environ Pollut, 10.1016/j.envpol.2019.06.004, 2019.*

*Liu, Y., Shen, H., Mu, J., Li, H., Chen, T., Yang, J., Jiang, Y., Zhu, Y., Meng, H., Dong, C., Wang, W., and Xue, L.: Formation of peroxyacetyl nitrate (PAN) and its impact on ozone production in the coastal atmosphere of Qingdao, North China, Sci Total Environ, 778, 146265, 10.1016/j.scitotenv.2021.146265, 2021.*

Line 99: "was attributed to the downwind region of the downtown (Xiamen island) with densely population " can be simplified to "was downwind of the densely populated downtown region (Xiamen island)"

**Response:** Thank you for your suggestions, we have simplified the description.

Line 100 in my opinion field observation should be plural "field observations were" not singular

**Response:** Thank you for your suggestions, we agree with your opinion and changed its format into plural.

Line 122 I have never come across the term "ultrasonic atmospherium" before. I believe the correct name would be "weather station with 2D sonic anemometer".

**Response:** Thank you for your suggestions. We are sorry for the wrong usage of the term, and the correct name you mentioned was right. Hence, we changed the name to "weather station with sonic anemometer".

Line 209 please avoid colloquial language "The wind directions in late spring and early autumn were messy due to the season switch."  More scientific "During the transition from spring to summer the wind direction fluctuated between … and … while during the transition from summer to autumn the wind direction fluctuated from … to …"

**Response:** Thank you for your suggestions. We have revised this description as "During the transition from spring to summer the wind direction fluctuated between northwest and southeast while during the transition from summer to autumn the wind direction fluctuated from southeast to northeast".

Line 210, respectively missing at the end of the sentence "The wind rose charts showed that the wind direction frequencies with relatively high wind speed in spring and autumn were southeast wind and northeast wind (Fig. S2), respectively." Also define "high" by inserting a number "(>… m/s)" in brackets.

**Response:** Thank you for your suggestions. We have revised this sentence as "The wind rose charts showed that the wind direction frequencies with relatively high wind speed ($>3$ m·s$^{-1}$) in spring and autumn were southeast wind and northeast wind (Fig. S2), respectively".

In general, the manuscript should be run through a grammar check software before resubmission. It is better to avoid colloquial language and indirect phrases. It is OK to be direct and use simple sentences.

**Response:** We have performed the grammar corrections of our manuscript through grammar check software, and we also invited native speakers in related fields to polish the manuscript. Thanks for your suggestion, simple sentences, and direct phrases make my point clear.

**(1) Response to Editor**

Comment on acp-2021-948

General remarks:

The authors present and extensive statistical analysis of a large dataset comprising an impressive number of parameters analysed. They use a model approach to assess the contribution of various pathways to PAN production. The he manuscript fits within the scope of ACP, however, I recommend some modifications prior to starting the public discussion. Overall, the manuscript would geratly benefit from another iteration of language editing as on several instances the wording is unclear and confusing.

**Response:** Thank you very much for your advice. We appreciate the opportunity to revise the paper. We have carefully addressed the comments and revised the manuscript accordingly.

The titel is not very clear - what is meant by 'Seasonal variation characteristic'? Consider a more concise wording.

**Response:** Thanks for your suggestion. We have revised the title of this manuscript as "Seasonal characteristics of atmospheric peroxyacetyl nitrate (PAN) in a coastal city of Southeast China: Explanatory factors and photochemical effects".

Throughout the manuscript the authors erroneously use althouh they are referring to mixing ratio(s) according to the units used.

This needs to be corrected.

**Response:** Thanks for your suggestion. We're sorry for the wrong usage of the word 'concentration(s)', and we have corrected it in the manuscript.

The description of the oberservations is very short and without a lot of further reading it is not possible to judge the quality of the data used. Please be more detailed in the description of the measurements. At minimum

a table of measurement precision and detection limits should be included in the supplement for all compounds whicht are discussed individually in the results sesctions. Please give details about the mentioned quality asssurance procedures applied.

**Response:** Thanks for your suggestion. We have improved it in the manuscript. More details about the methods and instruments have been added, as follows:

"PAN was monitored using a PAN analyzer (PANs-1000, Focused Photonics Inc., Hangzhou, CN) containing gas chromatography with electron capture detector (GC-ECD). During the observation period, multi-point standard curve calibration was conducted once a month, and single-point calibration was conducted every week, respectively. In the calibration mode, the Mass Flow Controller (MFC) controls the flow rate of NO, acetone and zero gas separately. The PAN standard gas is generated by the reaction of NO and acetone under ultraviolet light irradiation, and the sample is diluted to the required calibration mixing ratio for injection analysis. PAN was detected every 5 min and the detection limit was 50 pptv. The uncertainty and precision of PAN measurement were ±10% and 3%, respectively. Criteria air pollutants of $O_3$, CO, $SO_2$, and NOx, were monitored by using Thermo Instruments TEI 49i, 48i, 43i, and 42i (Thermo Fisher Scientific, Waltham, MA, USA), respectively. Particulate matters ($PM_{2.5}$) were monitored by oscillating microbalance with tapered element (TEOM1405, Thermo Scientific Corp., MA, US), and the uncertainty of the $PM_{2.5}$ measurement was ±20%, respectively. The meteorological parameters (i.e. wind speed (WS), wind direction (WD), pressure (P), air temperature (T), and relative humidity (RH)) were measured by an ultrasonic atmospherium (150WX, Airmar, USA). Ultraviolet radiation (UV) was determined by a UV radiometer (KIPP & ZONEN, SUV5 Smart UV Radiometer). HONO was monitored using an analyzer for Monitoring Aerosols and Gases in Ambient Air (MARGA, ADI 2080, Applikon Analytical B.V., the Netherlands). A gas chromatography-mass spectrometer (GC-FID/MS, TH-300B, Wuhan, CN) was used for monitoring the atmospheric VOCs with a 1-hour time resolution. The single-point calibration was performed every day at 23:00 with the standard mixtures of PAMS and TO15, and multi-point calibration was performed one month. The detection limits of the measured VOCs were in the range of 0.02 ppbv to 0.30 ppbv, and the measurement precision was ⩽10%. Photolysis frequencies including $J(O^1D)$, $J(NO_2)$, $J(HONO)$, $J(NO_3)$, $J(HCHO)$, and $J(H_2O_2)$ were analyzed by a photolysis spectrometer (PFS-100, Focused Photonics Inc., Hangzhou, China), and the uncertainty and detection limit of photolysis rates measurement were ±5% and around $1\times10^{-5}$, respectively. Table S1 shows the detailed uncertainty and detection limit of instruments for trace gas observation. A schedule was applied to operate and inspect the AEOS monitoring station regularly and strictly to ensure the validity of the data".

**Table S1. Detailed uncertainty and detection limit of instruments used for trace gas observation at the observation site.**

| Parameter | Experimental Technique | Uncertainty | Detection limit |
|---|---|---|---|
| PAN | PANs-1000, Focused Photonics Inc., Hangzhou, CN | ±10% | 50 pptv |
| $O_3$ | Model 49i, Thermo Fischer Scientific, USA | ±5% | 1 ppbv |
| NOx | Model 42i, Thermo Fischer Scientific, USA | ±10% | 0.5 ppbv |
| CO | Model 48i, Thermo Fischer Scientific, USA | ±5% | 40 ppbv |
| $SO_2$ | Model 43i, Thermo Fischer Scientific, USA | ±10% | 0.5 ppbv |
| VOCs | GC-FID/MS, TH-300B, Wuhan, CN | ±10% | 20-300 pptv |
| HONO | MARGA, ADI 2080, Applikon Analytical B.V., the Netherlands | ±20% | 50 pptv |

The conclusions section is underdeveloped and not very meaningful. What are the implications of the study regarding air quality monitoring (e. g. which key compounds should be measured) and air quality measures (regulation of activities/emissions). Are the results transferrable to other observation sites in different environments?

**Response:** Thanks for your suggestion. The results in our study indicate that the monitoring of PAN and its

precursors and the quantification of its impacts on $O_3$ formation have significant guidance on photochemical pollution control, and these findings can be applied to similar regions in relatively clean coastal cities. The scientific analysis methods used in our study provide a reference for the research on the formation mechanism of PAN and $O_3$ in other regions. We have revised the conclusion section in the manuscript.

"Field observation was continuously conducted in spring and autumn in a coastal city of Southeast China. We clarified the seasonal variations of PAN pollution, formation mechanisms, influencing factors and impacts on $O_3$ production. The average levels of PAN in autumn were lower than that in spring, while the $O_3$ showed the opposite characteristics. The multiple-factor GAM model showed that the key factors on PAN mixing ratio were UV, Ox, and T in spring, while Ox, TVOCs, T and $PM_{2.5}$ played important roles in PAN formation in autumn. The MCM model is an ideal tool to explore PAN photochemical formation and its key precursors at the species level and provides more relevant suggestions for reducing photochemical pollution. The controlling emissions of aromatics and alkenes with ≤5 carbons were benefit for PAN pollution mitigation, and carbonyl compounds especially acetaldehyde were dominant in the PAN production mechanism. PAN presented the inhibition or promotion effects on $O_3$ under different environmental conditions. The promotion effects of PAN on $O_3$ mainly happened during the period of 11:00-16:00 LT, most of which concentrated on PAN pollution episodes. According to the GAM analysis, the levels of ROx and UV were the main factors leading to the promotion effects in both seasons. Overall, PAN stimulated $O_3$ formation under high levels of UV, T and ROx in the coastal city. These results indicate that the monitoring of PAN and its precursors and the quantification of its impacts on $O_3$ formation have significant guidance on photochemical pollution control. The scientific analysis methods used in this study provide a reference for the research on the formation mechanism of PAN and $O_3$ in other regions".
* * *
Remarks on figures:

Figure captions are not very informative in the manuscript and the supplement. Please extend them.

**Response:** Thanks for your suggestion, we have extended the figure captions in the manuscript and the supplement.

Figures S2 and S3 lack label and units of the colorbar

**Response:** Thanks for your suggestion, we have added the lack label and the units in Figures S2 and S3 in the the supplement.

Figure S2 has no label of radial axis

**Response:** Thanks for your suggestion, we have added the lack label of radial axis in the Figures S2 in the supplement.

[Figure]

**Fig. S2. Wind direction frequency and wind speed plots in (a) spring and (b) autumn during the observation periods.**

Figures S7 and S8 are too small and at poor resolution

**Response:** Thanks for your suggestion, and we have changed to higher resolution and bigger figures.

[Figure]

**Fig. S7. Response curves (spring) of △P(O₃) to changes in (a) △ROx, (b) ultraviolet radiation (UV), (f) air temperature (T), (g) relative humidity (RH), and (h) wind speed (WS). The y-axis is the smoothing function values. For example, s(UV, df) shows the trend in △P(O₃) when UV changes, and the number of df is the degree of freedom. The x-axis is the influencing factor, and the shaded area around the solid red line indicates the 95% confidence interval of △P(O₃). The blue vertical short lines represent the concentration distribution characteristics of the explanatory variables (units: △ROx (molecules·cm⁻³), UV (W·m⁻²)).**

[Figure]

**Fig. S8.** Response curves (autumn) of $\triangle P(O_3)$ to changes in (a) $\triangle ROx$, (b) ultraviolet radiation (UV), (c) air temperature (T), (d) wind speed (WS). The y-axis is the smoothing function values. For example, s(UV, df) shows the trend in $\triangle P(O_3)$ when UV changes, and the number of df is the degree of freedom. The x-axis is the influencing factor, and the shaded area around the solid red line indicates the 95% confidence interval of $\triangle P(O_3)$. The blue vertical short lines represent the concentration distribution characteristics of the explanatory variables (units: $\triangle ROx$ (molecules·cm$^{-3}$), UV (W·m$^{-2}$), T (°C), WS (m·s$^{-1}$)).

Fig 3: Please explain how to read the amount of data indicator (dark blue vertical short lines).

**Response:** For a better understanding, we re-made the graph and added a scatter plot (like Fig. S7 and S8). It may be more appropriate to express the dark blue vertical short lines as the concentration distribution characteristics of the explanatory variables.

Fig 6: what does the central text annotation 'Autumn' refer to

**Res**ponse: The 'Autumn' in Fig. 6 refers to the relative incremental reactivity for major PAN precursor groups and specific species during the observation periods in autumn. We reformatted the pictures of Fig. 6, hoping to express the meaning more clearly.

[Figure]

**Fig. 6. The OBM-MCM calculated relative incremental reactivity (RIR) for major PAN precursor groups and specific species in (a) spring and (b) autumn during the daytime (06:00-17:00 LT).**

Fig 8: use higher resolution figure

**Response:** Thanks for your suggestion, we have changed a higher resolution figure.

[Figure]

**Fig. 8. Model-simulated (a) net O₃ production rate and O₃ budgets, (b) OH, HO₂ and RO₂ on the inhibition effect stages and promotion effect stages. Note: the white background parts represent the SC1 scenarios using the MCM mechanism, and the gray background parts represent the SC2 scenarios using the MCM mechanism with PAN chemistry disabled.**
* * *
Specific comments on text

Lines 22-24: What do the numbers in parenthese refer to? Do they have units?

**Response:** Thanks for your suggestion, and we're sorry for the unclear expressions. The numbers are F-values of GAM results, and F-value is a dimensionless statistic, which reflects the importance of the influencing factors. We have revised the sentence as "The F-values of GAM results reflecting the importance of the influencing factors showed that ultraviolet radiation (UV, F-value=60.64), Ox (Ox=$NO_2$+$O_3$, 57.65), and air temperature (T, 17.55) were the main contributors in the PAN pollution in spring, while the significant effects of Ox (58.45), total VOCs (TVOCs, 21.63) and T (20.46) were found in autumn".

L 30-31: The term 'promotion effect stages of PAN' is introduced later in the manuscript but here is totally unclear.

**Response:** Thanks for your suggestion, and we have revised the sentence as "The PAN promoting $O_3$ formation mainly occurred during the period of 11:00-16:00 (local time) when the favorable meteorological conditions (high UV and T) stimulated the photochemical reactions to offer ROx radicals, which accounted for 17% of the whole monitoring periods in spring and 31% in autumn".

L 43-44: I doubt that all precursors are produced by human activities only.

**Response:** Your understanding is correct, and we have revised the sentence as "PAN is generated through photochemical reactions of precursors emitted by human activities only".

L 64: Concentration values need to have a unit. Mixing ratio values (what's likely meant here) striclty spaking may not need a unit, but commonly ppb, nmol/mol or similar would be used.

**Response:** Thanks for your suggestion, and we have revised the sentence as "The mixing ratio of PAN in cities is higher than that in rural and remote areas, and that in background areas such as oceans and mountains can be as low as tens of pptv".

L 69: Some word mssing following 'suburban'.

**Response:** Thanks for your suggestion, and we have added the words "suburban regions".

L 97: What is meant by 'high dense vehicles'. Please reword.

**Response:** Thanks for your suggestion, and we have revised the words as "dense population and heavy traffic".

L 99-100: The term 'chosen' implies that the measurements were conducted over a loongre period and the two period of 53 were selected from this. Please motivate that choice.

**Response:** Thank you for your good comments and suggestions. As you said, our measurements were continuously conducted from March 15 to November 4, 2020. Exploring the photochemical pollution

mechanism in the southeast coastal area is our purpose of this research, which needs favorable and stable atmospheric environment for photochemical reaction. The photochemical pollution events mainly appeared during spring and autumn in Xiamen, so we preferred to choose the periods with relatively high $O_3$ and PAN levels, then screened and deleted some special circumstances, such as extreme weather conditions (typhoon, torrential rain, etc.) and instrument calibration. Finally, 53 days per season were chosen to study the characteristics of photochemical mechanism, and the field campaigns were conducted around ranged from 7 to 45 days in many studies (Liu et al., 2021a; Liu et al., 2021b), thus the 53 days per season that we chose were reasonable and representative. We have added the illustration of the choice in the revised manuscript, as follows:

"The photochemical pollution events mainly appeared during spring and autumn in Xiamen, and we preferred to choose the periods with relatively high $O_3$ and PAN levels, then the measured data of 53 days in each season was chosen after excluding some special circumstances, such as extreme synoptic situations and instrument calibration".

L 107: Please explain how a multi-point and a single-point calibratin can be perfromed at the same time. Give more details.

**Response:** Thank you for your suggestions. We're sorry for the confused expressions, and the multi-point and single-point calibration were not be performed at the same time. Based on the sensitivity testing experience of the PAN analyzer, it is necessary that the multi-point calibration is conducted once a month. We set the required calibration mixing ratio of 0.5, 1, 2, 3, 4, 5 and 6 ppbv for injection analysis to calibrate the instrument, which will spend 1-2 days and lead to that the data during multi-point calibration periods is unavailable. The single-point calibration (the setting mixing ratio was 2 ppbv) instrument baseline and sensitivity tested and calibrated weekly through, and the single-point calibration can help to find instrument instability under special circumstances. We have added the details about the calibration methods, as follows:

"During the observation period, multi-point standard curve calibration was conducted once a month, and single-point calibration was conducted every week, respectively. In the calibration mode of the PAN analyzer, the Mass Flow Controller (MFC) controls the flow rate of NO, acetone and zero gas separately. The PAN standard gas is generated by the reaction of NO and acetone under ultraviolet light irradiation, and the sample is diluted to the required calibration mixing ratio for injection analysis. PAN was detected every 5 min and the detection limit was 50 pptv. The uncertainty and precision of PAN measurement were ±10% and 3%, respectively".

Line 192ff: Why is there no mentioning of the pronounced contribution from the NNW sector here?

**Response:** Because it was the predominant wind direction we discussed relating to both wind speed and wind direction frequency, the contribution from the NNW sector was not the predominant wind direction. Figure S2 shows the wind direction frequency and wind speed plots in spring and autumn during the measurements. According to the wind rose plots, we found that the winds from the NNW sector had low wind speeds, while the wind speeds of winds from southeast (spring) and northeast (autumn) were much higher. Meanwhile, the direction of NNW is mainly rural residential and mountainous areas with less anthropogenic emissions, so that it is not the focus of this research. On the other hand, combined with Fig. S3, high $O_3$ and PAN values were concentrated on the wind direction of southeast and northeast, which proved the dominant wind direction again. Hence, there is no mentioning of the contribution from the NNW sector. We revised the sentence in the manuscript as follows:

"The wind directions in late spring and early autumn were messy due to the season switch. The wind rose charts showed that the wind direction frequencies with relatively high wind speed in spring and autumn were

southeast wind and northeast wind (Fig. S2). Although the frequency of northwest wind (NNW) also accounted for a certain proportion, the NNW speeds were generally slow, and the direction of the NNW was mainly rural residential and mountainous areas with less anthropogenic emissions, so that it was not the focus of this research".

L 197/198: It is not evident why and by which mechanism the mentioned parameters should in general favour an accumulation of pollutants. Please explain.

**Response:** Thank you for your suggestions. Xiamen is under the East Asian monsoon control, belonging to the subtropical marine climate. Autumn is controlled by the west pacific subtropical high, carrying favorable photochemical reaction conditions (high temperature, low RH, and stagnant weather conditions) and encouraging the formation and accumulation of $O_3$ in the southeast coastal area (Wu et al., 2019; Liu et al., 2021c), which is why there is a lot of photochemical pollution in autumn. Our previous study showed that Xiamen is influenced by WPSH >9 months every year and the transition to winter season was from September to November generally. Hence, we are used to considering that the weather condition in autumn is good for the accumulation of pollutants. We strongly agree with your question and revised the expression, as follows:

"These meteorological conditions carried by the WPSH (high T, low RH, and stagnant weather conditions) were conducive to the photochemical reaction and accumulation of air pollutants in autumn (Wu et al., 2019; Xia et al., 2021)".

L 231: why would PA+NO be a loss pathway of PAN?

**Response:** We're sorry for the confused expressions. PAN can be produced only through the reaction of $PA+NO_2$ (Liu et al., 2021a; Xue et al., 2014). Hence, the production and sink of PA can represent that of the PAN mechanism indirectly. The main and direct PAN destruction is thermal decomposition, and the indirect sinks of PAN were the reactions of PA with NO, $HO_2$, and $RO_2$ (Zeng et al., 2019). Hence, we revised the expression, as follows:

"As we all know, the reaction of PA+NO is one of the most important loss pathways of PA, suggesting the fact that NO consumed PAN indirectly".

L 277: why would the consumption of precursors of CO, VOC and NO matter here?

**Response:** Through the analysis of the distribution characteristics of the monitoring values, the characteristics of pollution and pollution sources can be initially revealed. Here should be CO, NOx and VOCs. Because CO, NOx and TVOCs are the precursors of PAN and $O_3$, which can directly affect photochemical reactions. The precursors of CO, NOx and VOCs were consumed to produce photochemical products during the daytime, and were accumulated during the nighttime without the solar radiation. PAN and $O_3$ are secondary pollutants depending on both the levels of precursors and photochemical reaction rates. Therefore, the trends of PAN and $O_3$ were opposite to that of their precursors in our observation site. In our study, NO and $NO_2$ showed similar trend and reached the maximum at the same time, indicating the high atmospheric oxidation capacity in Xiamen, thereby, NO was quickly oxidized to $NO_2$ (in our previous study, atmospheric oxidation capacity was indeed higher than that in some urban cities, such as Shanghai, Hong Kong and Qingdao (Liu et al., 2021c)). CO, NOx and TVOCs showed the highest levels at around 08:00 LT due to the nighttime accumulation and vehicle exhaust, but showed relatively low levels during the other periods of the daytime, emphasizing the importance of anthropogenic sources and weather conditions in our observation site. Hence, the consumption of precursors of CO, VOC and NOx were matter here. The revised in the manuscript were

shown, as follows:

"CO, NOx and TVOCs showed the highest levels at around 08:00 LT due to the nighttime accumulation and vehicle exhaust, but showed relatively low levels during the daytime, emphasizing the importance of frequent human activities and weather conditions".

L 302: shouldn't it be PA here (rather than PAN)

**Response:** Just as the answer to the question 'L 231', PAN can be produced only through the reaction of PA+$NO_2$ (Liu et al., 2021a; Xue et al., 2014). Hence, the production and sink of PA can represent that of the PAN mechanism indirectly. In many studies, they expressed that "production and loss of PA radical (hence PAN)" (Liu et al., 2021a; Xue et al., 2014; Zeng et al., 2019). We revised the expression, as follows:

"Both the PA (hence PAN) production and destruction rates during episodes were 1.80 times higher than those during non-episodes".

L 374ff: What do you mean by 'opening and closing PAN photochemistry'? Does it mena there were too model runs with PAN photochemistry included in only on? Do the delta values in the following refer to the difference between these to model runs? Explain in more detail and clearer wording.

**Response:** Thank you for your suggestions. We have changed the expression and added a detailed description, hoping to help readers understand what we mean more clearly. The detailed modifications are as follows:

"To quantify the changes of $O_3$ in response to PAN chemistry in the coastal city, two parallel scenarios (SC1 and SC2) were conducted based on the OBM model. The SC1 was the base scenario putting all detected data (i.e. VOCs, trace gases and meteorological parameters) into the model with all reaction pathways (as the description in Section 2.2), and the SC2 disabled the PAN chemistry, which is the only difference between SC2 and SC1. Figure 7 shows the differences of $O_3$ net production rates $\triangle P(O_3)$, $\triangle OH$, $\triangle HO_2$, $\triangle RO_2$, $\triangle NO$ and $\triangle NO_2$ between the SC1 and the SC2".

**(2) Response to Editor**

**Comments to the author**:
thank you for submission of the revised version of your manuscript on PAN measurements. In general I am happy with your reponse to the reviewers and with your changes to manuscript, but there are three minor changes I ask you to to include prior to final acceptance of the draft:

**Response:** Thanks for your valuable comments and positive feedback. We have corrected this manuscript carefully according to your suggestion.

-- From your reply to Reviewer #1 you added figures T1 and T2 to the supplemnets. I thinks these figures are very useful. Please add their captions to the introductory part of the supplementary document. In addition, I suggest to use the same axis scaling for the spring and autumn plots to facilitate comparison. In particular, I was confused by the different histogram binning used in panel (c) of the figures.

**Response:** Thank you very much for your advice. We have added their captions to the introductory part of the supplementary document. We have used the same axis scaling for the spring and autumn plots, and also

changed the figures caption of T1 and T2 as S1 and S2. The detailed corrections are as follows:

[Figure]

**Fig. S1 Residual test results of the Generalized Additive Model (GAM) in spring.**

[Figure]

**Fig. S2 Residual test results of the Generalized Additive Model (GAM) in autumn.**

-- the changes you made to Lines 251-255 (track change document) do not improve the manuscript. Words like "values", "abundance" and "levels" are vague and do not refer to physical, measurable quantities. Please return to the intial use of "mixing ratio". Your reply to the reviewers comment is sufficient, but I do not agree with the wording changes.

**Response:** Thank you very much for your suggestions, we have returned to the initial use of "mixing ratio, as follows:

"Based on the above analysis, we found that the photochemical reactions were still intense and even stronger under the low precursor mixing ratios. Although the precursor mixing ratios of PAN and $O_3$ in spring were significantly higher than those in autumn (P<0.01), the PAN mixing ratios in autumn were comparable to those in spring, while the $O_3$ mixing ratios in autumn were much higher than those in spring. Therefore, it is very necessary to furtherly explore the key influencing factors and their formation mechanisms".

-- In your reply to Reviewer #2 you make a statement on the selection of compounds from the full set of 106 substances identified in your analysis. Please add a corresponding sentence (around Lines 127-129 of the track change manuscript) with a reference to table S2 and a mentioning of the lack of calibration gases for some compounds.

**Response:** Thank you very much for your suggestions, we have corrected this part accordingly.

"Nine compounds (Acetaldehyde, Propanal, Crotonaldehyde, Methacrolein, n-butanal, Benzaldehyde, Valeraldehyde, m-Tolualdehyde, Hexanal) could not be determined due to lack of aldehyde and ketone calibration gases, and Table S2 showed all VOCs compounds that we used in the OBM model".